# Cerebellar implementation of movement sequences through feedback

**Andrei Khilkevich[1], Juan Zambrano[1], Molly-Marie Richards[1], Michael Dean Mauk[1,2]\***

[1]Center for Learning and Memory, The University of Texas at Austin, Austin, United States; [2]Department of Neuroscience, The University of Texas at Austin, Austin, United States

**Abstract** Most movements are not unitary, but are comprised of sequences. Although patients with cerebellar pathology display severe deficits in the execution and learning of sequences (Doyon et al., 1997; Shin and Ivry, 2003), most of our understanding of cerebellar mechanisms has come from analyses of single component movements. Eyelid conditioning is a cerebellar-mediated behavior that provides the ability to control and restrict inputs to the cerebellum through stimulation of mossy fibers. We utilized this advantage to test directly how the cerebellum can learn a sequence of inter-connected movement components in rabbits. We show that the feedback signals from one component are sufficient to serve as a cue for the next component in the sequence. In vivo recordings from Purkinje cells demonstrated that all components of the sequence were encoded similarly by cerebellar cortex. These results provide a simple yet general framework for how the cerebellum can use simple associate learning processes to chain together a sequence of appropriately timed responses.
DOI: https://doi.org/10.7554/eLife.37443.001

## Introduction

Most movements are comprised of sequences. From the complex routines that gymnasts perform to intricate dance numbers to simply reaching for an object, our movements are comprised of sequences of movements that are learned through practice. The cerebellum has been long implicated in learning and execution of accurate movements (*Doyon et al., 1997*; *Shin and Ivry, 2003*; *Lehéricy et al., 2005*; *Krupa et al., 1993*; *Lisberger, 1994*; *Diener and Dichgans, 1992*). Movement sequences as well as multi-joint movements are particularly sensitive to cerebellar dysfunction (*Shin and Ivry, 2003*; *Diener and Dichgans, 1992*; *Doyon et al., 2002*). For example, one of the hallmark deficits of cerebellar pathology is dysdiadochokinesia (*Diener and Dichgans, 1992*) – an inability to perform a rapid alternating sequence of movements. Patients with cerebellar lesions display severe deficits (*Doyon et al., 1997*) in sequence learning or are unable to learn sequences at all (*Shin and Ivry, 2003*), even with modest impairments in learning of single-component, directly cued movements (*Spencer and Ivry, 2009*). Studies of sequence learning in nonhuman primates (*Desmurget and Turner, 2010*; *Rünger et al., 2013*) demonstrated that with repeated training components of movement in the sequence start to be initiated predictively, before the arrival of sensory cue. Since one of the fundamental properties of cerebellar learning is the ability to learn a predictive response (*Marr, 1969*; *Bastian, 2006*; *Shadmehr et al., 2010*; *Therrien and Bastian, 2015*), these results also indirectly imply a strong cerebellar contribution to sequence learning.

Although results of these studies suggest cerebellar involvement in the learning and execution of movement sequences, most of what we know about cerebellar mechanisms of learning comes from studies utilizing single-component movements. These include adaptation of the vestibule-ocular reflex, adaptation of saccadic and smooth pursuit eye movements and conditioning of eyelid

**\*For correspondence:**
mauk@utexas.edu

**Competing interests:** The authors declare that no competing interests exist.

**eLife digest** Imagine a gymnastics competition in which participants take turns to cartwheel and somersault across the floor. The routines on display comprise sequences of precisely timed movements learned through practice. This is also true for many of the actions we perform every day, such as reaching for a cup of coffee. A region of the brain called the cerebellum helps us learn sequences of movements. But how does it do this?

To find out, Khilkevich et al. came up with a new version of an old experiment. Rabbits were first trained to blink their eye in response to a specific external cue. This type of learning, called associative learning, has been shown before in the cerebellum. But Khilkevich et al. wondered whether the cerebellum could also use internal feedback signals from the eyeblink as a cue to learn the next movement? If so, this might explain how the cerebellum can chain movements together in a sequence.

As predicted, Khilkevich et al. found that rabbits could learn to blink their eye in response to an initial signal, and then blink again in response to the first blink. Control experiments confirmed that the second eyeblink was coupled to the first, and not to the original cue. Moreover, on many trials the rabbits showed a third and even fourth eyeblink. This is because feedback signals from the first, second or third blink were the same. Thus, the feedback signals from the first blink triggered the second blink, feedback from the second triggered the third, and so forth. Rabbits could also learn to use a blink of the left eye as a cue for a blink of the right eye. Similar patterns of neuronal activity accompanied each blink, suggesting that the same mechanism generated them all.

The cerebellum can thus use feedback from one movement as a cue to learn the proper timing of the next movement in a sequence. A key question is whether this mechanism of sequence learning extends beyond movement. The cerebellum has extensive connections to the brain's outer layer, the cortex, including many areas involved in cognition. Future experiments should test whether the cerebellum might help guide sequences of cortical activity during cognitive tasks.

DOI: https://doi.org/10.7554/eLife.37443.002

responses. In principle, the cerebellar mechanisms underlying movement sequences could be different from, or at least somewhat different from those mediating single-component movements. If so, these mechanisms are largely unknown. In contrast, we tested the hypothesis that a simple elaboration of cerebellar mechanisms that mediate learning single-component movements is sufficient to explain cerebellar learning and implementation of movement sequences.

Three possible (but not mutually exclusive) ways of implementing cerebellar learning of movement sequences are illustrated in *Figure 1*. With the first possibility a specific external cue is associated with a specific movement component in the sequence (*Figure 1A*). The second option is a variant of the first one. Here only single external cue is present, but it persists in time through the whole sequence, so that different movement components are elicited by signals associated with different times during the cue (*Figure 1B*). Finally, *Figure 1C* illustrates a possibility where feedback signals from one movement component are used to learn the next component. The design of most experiments does not permit distinguishing between these possibilities. For example, a number of studies (*Choi and Moore, 2003*; *Moore and Choi, 1997*; *Freeman et al., 2003*; *Halverson et al., 2015*; *Jirenhed et al., 2017*) used eyelid conditioning to train subjects to respond with a sequence of two eyelid responses. In these experiments, however, the external cue either explicitly extended through the whole sequence, or involved an auditory signal that can elicit persistent activity in working-memory areas such as mPFC (*Siegel et al., 2012*), precluding the possibility of determining which type of sequence learning applies (*Figure 1B* and *Figure 1C* are both possible). We utilized the practical and conceptual advantages of eyelid conditioning to test explicitly the sufficiency of feedback signal(s) (FS) for sequence learning (*Figure 1C*). We demonstrate directly that the cerebellum can learn to chain together a sequence of inter-connected movement components by using FS from one component to serve as a cue for the next component in the sequence (*Figure 1C*).

In a standard eyelid conditioning experiment, activation of mossy fiber inputs by a sensory stimulus is the cue the cerebellum uses to learn a predictive conditioned eyelid response (CR) (*Figure 1C*, first movement). A sensory stimulus can be replaced by direct electrical stimulation of mossy fibers,

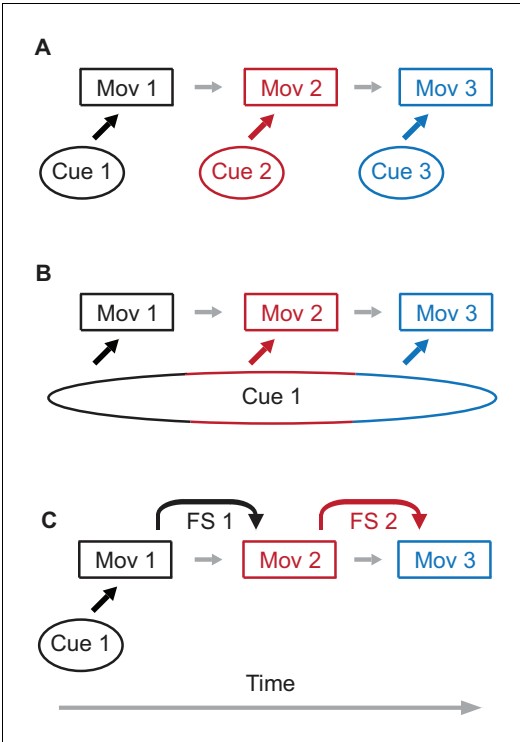

**Figure 1.** Possible mechanisms of sequence learning. (A) A sequence of three movement components is learned through association of each movement component with a separate external cue. (B) A single cue is present, but the duration of the cue persists through all three components of the sequence. In this case, different time epochs of the cue serve as effective separate cues from panel (A). (C) Only the first movement component is learned from the external cue. For the following movement components, feedback signals (FS) from a preceding movement are used to learn the next movement component, assembling the sequence of movements.
DOI: https://doi.org/10.7554/eLife.37443.003

which is an equally effective cue to support learning (*Steinmetz et al., 1985*), but has the advantage of restricting inputs only to the cerebellum. We employed electrical stimulation of mossy fibers as a cue to learn a single-component response. We then arranged the timing of stimuli in subsequent training so that the second component in the sequence could not be supported by the mossy fiber stimulation eliciting the first component and thus, only option shown in *Figure 1C* would be possible. We demonstrate robust learning of later components, including sequences of movements produced by the same muscle and sequences with different muscle groups used for the different components. A variety of control experiments showed how the presence and timing of the later components of responses were coupled to the first component and not to the mossy fiber stimulation cue that initiated the first component. In vivo recordings from cerebellar cortex showed that Purkinje cell activity relates as strongly to the latter components of sequences as it has been shown to relate to single-component eyelid responses. Together, these results show how the well-characterized cerebellar mechanisms of learning single movement components can be extended to learning of movement sequences and provide a general framework for how the cerebellum can use feedback signals to learn to chain together appropriately timed responses to produce a movement sequence.

## Results

### Training an ipsilateral sequence of CRs

To test the hypothesis that the cerebellum can use feedback signals from a movement as the cue for the next movement in a sequence, we started by training rabbits using electrical stimulation of mossy fibers as a cue (to which we refer as the conditioned stimulus or CS). All subjects (New Zealand albino rabbits) were initially trained by pairing a mossy fiber stimulation CS with a reinforcing unconditioned stimulus (US, electrical stimulation of the skin near the eye). The inter-stimulus interval (ISI) between CS onset and US onset was 500 ms. As has been previously demonstrated, this training yields robust and well-timed conditioned eyelid responses in response to the mossy fiber CS (*Steinmetz et al., 1985*; *Kalmbach et al., 2011*). Example conditioned eyelid responses are shown in *Figure 2A*. Use of mossy fiber stimulation as the CS ensured that the CS was restricted to the cerebellum and did not propagate to areas that could provide a delayed secondary input to the cerebellum (*Siegel and Mauk, 2013*; *Halverson et al., 2010*), which can be the case for auditory stimuli commonly used in eyelid conditioning. Moreover, the control over CS duration allowed control over the time gap between CS offset and the US.

Once subjects reached robust responding to the mossy fiber CS training was switched to protocols teaching a sequence of CRs. The design of an ipsilateral sequence training protocol is illustrated in *Figure 2B*. Training trials involved presentation of the same mossy fiber stimulation CS (500 ms long) used during initial training, but now, depending on the amplitude of the CR, the US was

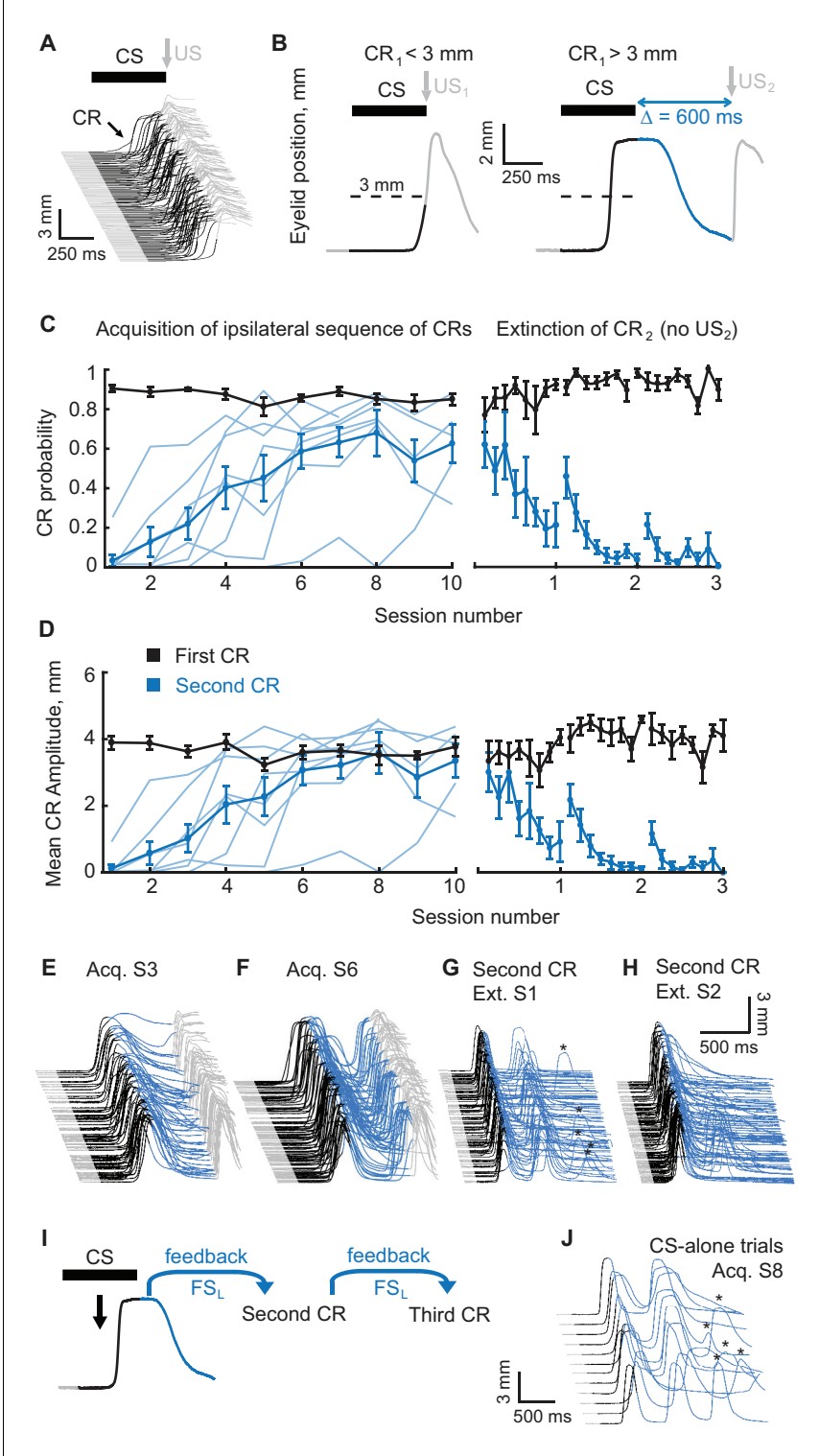

**Figure 2.** Acquisition and extinction of an ipsilateral sequence of CRs. (**A**) The waterfall plot shows eyelid position as a function of time on each trial with trials from one training session arranged chronologically from bottom to top. Upward deflection indicates closure of the eyelid. The black portion of each trace indicates the CS duration. Predictive eyelid closures beginning during the CS and before the US are cerebellar-driven conditioned responses. Reflexive (unconditioned) eyelid responses occurring after the US are not mediated by the cerebellum (light grey portion of traces). (**B**) A schematic representation of the ipsilateral sequence training protocol. Left panel shows an example trial with first CR amplitude smaller than the 3 mm target (dotted line). In these instances

*Figure 2 continued on next page*

*Figure 2 continued*

the US$_1$ is delivered. Right panel shows a trial with first CR amplitude larger than 3 mm. In these instances the US$_1$ is omitted and the US$_2$ is delivered. As before, the duration of the mossy fiber stimulation CS is shown in black, the interval between CS offset and the US$_2$ is shown in blue, grey portions of the sweep are the pre-CS and post US$_2$ periods. (**C**) CR probability as a function of session number. Left panel shows acquisition curves of second CR in ipsilateral sequences. The probability of first CRs is shown in black, the probability of second CRs in blue. Thin lines represent individual subjects, whereas thick lines show group averages. The right panel shows data from three sessions of second CR extinction. Data in extinction sessions was broken down into eight equal portions to evaluate the time course of extinction through the session. (**D**) Same as (**C**), but for average amplitude of CRs in ipsilateral sequence training. (**E–H**) Example raw data from acquisition and extinction sessions of ipsilateral sequence of CRs. In all cases only trials with first CR amplitude larger than 3 mm are shown. (**I**) Schematic of rationale why subjects trained at ipsilateral sequence of CRs should produce a third CR on CS-alone trials without it being explicitly reinforced during training. (**J**) Example CS-alone trials from late acquisition session. The third CRs in a sequence are indicated by asterisks above.

DOI: https://doi.org/10.7554/eLife.37443.004

The following source data and figure supplement are available for figure 2:

**Source data 1.** Behavioral data during acquisition of ipsilateral sequence of CRs.

DOI: https://doi.org/10.7554/eLife.37443.006

**Figure supplement 1.** Timing, acquisition and extinction of CRs in ipsilateral sequence training.

DOI: https://doi.org/10.7554/eLife.37443.005

---

presented at one of two different times. If the CR amplitude was below the target amplitude (3 mm, half of full closure) the US was presented as normal at CS offset (designated US$_1$). When the CR amplitude was at or above the target value at the scheduled US$_1$ time, the US was instead presented 600 ms after CS offset (designated as US$_2$). The purpose of the US$_1$ trials was to ensure continued robust responding to the mossy fiber CS, whereas the purpose of the US$_2$ trials was to train a second component of the movement. With this procedure we could then test whether the expression of the first CR could serve as a signal for the cerebellum to learn the second CR. A 600 ms gap between the offset of the mossy fiber stimulation CS and the presentation of US$_2$ was used based on previous findings that a temporal gap larger than 400 ms between the offset of a mossy fiber stimulation CS and the onset of a US does not support cerebellar learning of eyelid CRs (*Kalmbach et al., 2010a*). Given these factors, the ability to learn a second CR elicited near the time of US$_2$ would suggest that the feedback information about the first CR is sufficient for the cerebellum to use as a new 'CS' to learn the subsequent CR.

All subjects (N = 8) successfully acquired a sequence of ipsilateral CRs (*Figure 2C–F*) with the timing of the second peak appropriate for the time at which US$_2$ was presented (*Figure 2—figure supplement 1A*). The probability of the second CR, defined as the fraction of trials with first CR where second CR was present too, and second CR amplitude grew monotonically over several sessions of training (*Figure 2C and D* respectively, blue lines, one-way ANOVA, $F_{(9,63)} = 6.44$, $p<10^{-6}$; $F_{(9,63)} = 7.16$, $p<10^{-7}$ for second CR probability and amplitude respectively), and eventually reached asymptotic value, paralleling typical acquisition curves in eyelid conditioning.

Cerebellar learning is associative. As a consequence, presentation of trials with omitted US diminishes CRs in a process called extinction. As a test of whether the second CRs are associatively learned, we tested whether they can be selectively extinguished like normal CRs. Over three extinction sessions (*Figure 2C,D,G,H*), the US$_1$ was still presented according to the rule described above, but US$_2$ was omitted. If second CRs are learned associatively, the absence of reinforcing US$_2$ should lead to their gradual extinction. As expected, the probability and amplitude of the second CRs (blue lines) monotonically decreased (One-way ANOVA, $F_{(23,120)} = 4.96$, $p<10^{-9}$; $F_{(23,120)} = 4.92$, $p<10^{-9}$ for second CR probability and amplitude respectively) during extinction sessions, without any effect on the probability and amplitude of the first CRs (black lines) (One-way ANOVA, $F_{(23,120)} = 1.45$, $p=0.10$, $F_{(23,120)} = 1.35$, $p=0.15$ for first CR probability and amplitude). Similar to extinction in conventional eyelid conditioning protocols, spontaneous recovery (*Weidemann and Kehoe, 2004*; *Thanellou and Green, 2011*; *Ohyama et al., 2010*) of the second CRs was present at the beginning of the second and third extinction sessions (paired Wilcoxon signed rank test, p=0.023 for second CR probability, p=0.032 for second CR amplitude). Since the number of trials with first CR amplitude

larger than the 3 mm target was variable between sessions and animals, the same data plotted as a function of total trial number is shown in *Figure 2—figure supplement 1C,D*. Plotted this way, the acquisition and extinction rates appear similar to what was observed in conventional eyelid conditioning training protocols. These observations are consistent with the notion that the second components in the sequence are acquired through standard cerebellar-dependent associative learning.

The primary hypothesis we sought to test was that the cerebellum uses a feedback signal (FS) of some sort about the first CR as a separate 'CS' to learn the second CR (*Figure 2I*). Such a FS should occur anytime there is a CR, whether the CR is elicited by the mossy fiber stimulation CS or not. This suggests the prediction for ipsilateral sequence of CRs, illustrated in *Figure 2I*. When a well-trained subject is presented with CS-alone trials, the mossy fiber stimulation CS will elicit the first CR, then a $FS_L$ from the first CR will elicit the second CR. Because the second CR is produced by the same muscles of left eyelid and is driven by likely the same neurons in cerebellar nuclei, then if the second CR is large enough, its FS should be similar to the $FS_L$ from the first CR. Since the cerebellum already learned to associate $FS_L$ with upcoming $US_2$, it should produce a predictive CR with appropriate timing. Thus, we would expect to see a third (and perhaps fourth and fifth, etc.) CRs on CS-alone trials, though subjects were never explicitly trained to produce them. This is indeed what we observed (*Figure 2G,J*). Third and following CRs appeared in late acquisition sessions once robust second CR performance was established and disappeared with the extinction of the second CRs (*Figure 2J,H*). On 36% of CS-alone trials where there was a second CR, there was also a third CR. The relative time between peaks of second and third CRs was similar to the time between peaks of first and second CRs (*Figure 2—figure supplement 1B*). Though third CRs were never explicitly reinforced, they remained through late acquisition sessions and subsequent control experiments, as long as second CRs also remained. This observation is consistent with the idea that both second and third CR share the same 'CS' as $FS_L$ and reinforcement of second CRs on a portion of trials was sufficient to maintain both second and third CRs. Because we recorded eyelid position for 2500 ms on each trial, we were only able to observe up to four CRs with ipsilateral sequence training. The existence of third CRs is inconsistent with the idea that the original mossy fiber stimulation CS is driving the expression of the second (and later) CRs.

## Training contralateral sequence of CRs

Complex movements typically involve more than one muscle group and often bilateral coordination (*Castiello et al., 1993*; *Kelso et al., 1979*). We therefore asked whether this training protocol can support a sequence where a left eyelid CR would be followed by a right eyelid CR (*Figure 3A*). As before, subjects were initially trained with mossy fiber stimulation CS to produce left eyelid CRs at ISI 500 ms. After successful acquisition, subjects were switched to a contralateral CRs sequence protocol. Here again, if the left eyelid CR amplitude was lower than the target (3 mm, half-sized CR), $US_L$ was presented to the left eye to maintain robust responding of the left eyelid CRs. If however, the amplitude of the left CR was higher than the target, $US_L$ was omitted and $US_R$ was presented to the right eye. During initial acquisition, the interval between CS offset and $US_R$ was typically 400 ms (N = 4), but for some subjects was 300 ms (N = 1) or 500 ms (N = 1). We chose to use a shorter duration of the gap interval compared to ipsilateral sequence training, because the pilot data showed that most subjects were unable to learn a contralateral sequence with 600 ms gap interval from naïve right eyelid state. In this situation, however, there is less concern about the ability of the mossy fiber stimulation CS to drive the right eyelid CR, since the mossy fiber stimulation CS was delivered through electrodes implanted in the left middle cerebellar peduncle. The only way that CS could propagate to the right cerebellar hemisphere was by antidromic activation of neurons in pontine nucleus that have bilateral axons projections. Since the number of such neurons is extremely low (*Serapide et al., 2002*; *Tan and Gerrits, 1992*; *Kratochwil et al., 2017*), it is unlikely that mossy fiber stimulation CS could support acquisition of right eyelid CRs. Several control experiments described later confirm this notion.

All subjects (N = 6) successfully acquired a contralateral sequence of CRs where a left eyelid response was followed by a right eyelid response (*Figure 3 B–E*, $F_{(9,46)}$ = 14.28, p<$10^{-9}$; $F_{(9,46)}$ = 9.05, p<$10^{-6}$ for right CR probability and amplitude respectively). Right eyelid CRs were adaptively timed, peaking near the time of $US_R$ delivery and thus corresponding to the gap interval used during training (*Figure 3—figure supplement 1A,B*). Unlike the ipsilateral training protocol, here there were no third CRs produced by either left or right eyelid. Assuming that all but the first (left) CRs are

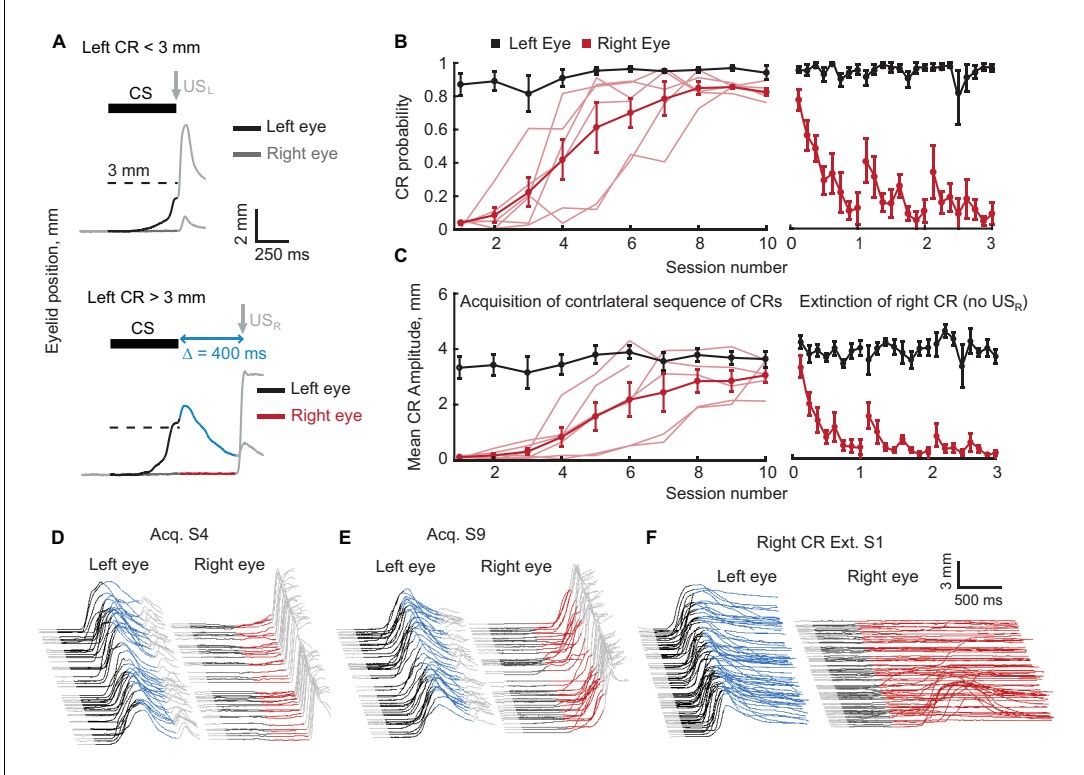

**Figure 3.** Acquisition and extinction of contralateral sequences of CRs. (**A**) A schematic representation of the contralateral sequence training protocol. Top panel shows an example trial with left eye CR amplitude smaller than target 3 mm value (dotted line) and where $US_L$ is delivered to the left eye. The bottom panel shows a trial with left CR amplitude larger than 3 mm. In these instances, $US_L$ is omitted and $US_R$ is delivered to the right eye. Color-coding of left eyelid position is the same as in *Figure 1*. For right eyelid position CS duration is indicated by a dark grey color, the interval between CS offset and $US_R$ is shown in red. (**B**) CR probability as a function of session number. Left panel shows acquisition curves of right eye CR in contralateral sequence. Probability of left eye CR is shown in black, right eye CR – in red. Thin lines represent individual subjects, thick lines – group averages. Right panel shows data from three sessions of right eye CR extinction. Data in each session were broken down into eight equal portions to evaluate the time profile of extinction through the session. (**C**) Same as (**B**), but for average amplitude of CRs in ipsilateral sequence. (**D–F**) Examples of acquisition and extinction sessions of contralateral sequence of CRs. For each session left eye responses are shown on the left, right eye responses – on the right. In all cases only trials with left eye CR amplitude larger than 3 mm are shown.

DOI: https://doi.org/10.7554/eLife.37443.007

The following source data and figure supplement are available for figure 3:

**Source data 1.** Behavioral data during acquisition of contralateral sequence of CRs.

DOI: https://doi.org/10.7554/eLife.37443.009

**Figure supplement 1.** Timing, acquisition and extinction of CRs in contralateral sequence training.

DOI: https://doi.org/10.7554/eLife.37443.008

driven by FS, and assuming that $FS_L$ from left eyelid CRs and $FS_R$ from the right eyelid CRs are different signals, this is the expected result. In this protocol the $FS_L$ are paired with a right eyelid $US_R$ and the $FS_R$ are never paired with a US. Thus there should only be a left eyelid CR driven by the mossy fiber stimulation CS and a right eyelid CR driven by $FS_L$ from left eyelid CR. Through 86 sessions with contralateral sequence protocol from 6 subjects, responses following right eyelid CR were not observed for either the left or right eyelid (Data not shown).

As with the ipsilateral sequence protocol, we verified the associative nature of right eyelid CRs by performing three extinction sessions of right eyelid CRs (red lines in *Figure 3B,C,F*, *Figure 3—figure supplement 1C,D*, One-way ANOVA, $F_{(23,96)} = 4.04$, $p<10^{-7}$; $F_{(23,96)} = 6.89$, $p<10^{-12}$ for right CR probability and amplitude respectively). Extinction of right eyelid CRs did not influence the performance of preceding left eyelid CRs (One-way ANOVA, $F_{(23,96)} = 0.96$, $p=0.52$; $F_{(23,96)} = 0.82$, $p=0.70$ for left eyelid CR probability and amplitude respectively). Spontaneous recovery of right eyelid CRs at the beginning of second and third extinction sessions was present here as well (paired Wilcoxon

signed rank test, p=0.055 for right CR probability, p=0.020 for right CR amplitude). Following extinction sessions, subjects were switched to contralateral training protocol with a longer interval between CS offset and $US_R$ (500 ms, transition not shown). In the analysis of contralateral sequence data, data produced on sessions with different gap intervals are indicated with different colors and corresponding legend, if applicable.

The hypothesis that the cerebellum generates the later CRs using FS from the first CR gives rise to several testable predictions. We designed two control experiments and performed additional analyses on training session data as a formulation of these predictions to test relatively directly whether the second CR in a sequence (second left eyelid CR in ipsilateral protocol and right eyelid CR in contralateral protocol) is driven by FS from the first CR and not by the mossy fiber stimulation CS itself.

## Prediction 1: extinction of the first CR in the sequence should lead to extinction of the second CR

The design of the first control experiment is based on a straightforward prediction – the absence or a decline in the first CRs should also lead to an absent or diminished second CRs. We applied an extinction protocol designed to promote extinction of the first CRs and tested the effects on the second CRs. The setup of this experiment is shown in *Figure 4A,D*. Regardless of first CR amplitude, on these sessions the second CR was always reinforced with $US_2$ (or $US_R$ in the contralateral sequence), while $US_1$ ($US_L$) was never delivered. A decline of the second CRs despite the presence of reinforcing $US_2$ (or $US_R$), would indicate that expression of the second CRs requires the expression of the first CR in the sequence. *Figure 4B,E* shows CRs amplitudes is a sequence as a function of block number. Since extinction typically took longer for subjects trained with the ipsilateral sequence, most required two extinction sessions. The same type of plot for CRs probability is shown in *Figure 4—figure supplement 1A,B*. For subjects trained in either the ipsilateral or contralateral sequences, as the amplitude and probability of left eyelid CR decreased, so did the amplitude of the second left eyelid CR in ipsilateral sequence (r = 0.86, p<$10^{-13}$); or right eyelid CRs in contralateral sequence (r = 0.80, p<$10^{-12}$). To quantify this effect further we evaluated the probability of the second CRs conditioned on first CR amplitude. For that, we first plotted second CR amplitude versus first CR amplitude (*Figure 4—figure supplement 1C,D*, each dot represents a single trial from this control experiment). We then used that data to calculate the probability of the second CR conditioned on first CR amplitude (*Figure 4C,F*). For both ipsilateral and contralateral sequence, the probability of observing second CRs decayed with decrease in first CR amplitude (Chi-square analysis, $\chi^2(6, 836)=60.90$, p<$10^{-11}$, $\chi^2(6, 572)=92.7$, p<$10^{-18}$, for ipsilateral and contralateral sequence protocols respectively). On trials without the first left eyelid CR, the probability of second CRs was negligible (probability of second left eyelid CR = 0.026 ± 0.009 for ipsilateral sequence protocol, probability of right eyelid CR = 0.023 ± 0.01 for contralateral sequence protocol). These data show that the presence of a first CR is required for the expression of a second CR.

## Prediction 2: second CR does not depend on cue that drives the first CR

If a FS is the cue that drives the second CR, then it should not be necessary for the first CR to be elicited by the original training CS. Any CR produced by that eyelid, however it was elicited, should serve as an effective source of a FS to produce a second CR. The schematic of the second control experiment that we used to test this prediction is shown in *Figure 5A,E*. We started by training subjects to produce left eyelid CRs (ISI 500 ms) with two different types of CS: CS1 and CS2. Electrical stimulation of mossy fibers was always used as CS1, CS2 was either a 500 ms mossy fibers stimulation delivered through a separate electrode (N = 5, 2 subjects for ipsilateral sequence and 3 subjects for contralateral sequence) or a 500 ms 1 kHz tone (N = 5, 3 subjects for ipsilateral sequence and 2 subjects for contralateral sequence). At the end of this pre-training each subject elicited robust CRs to the presentation of either CS1 or on separate trials to CS2. Then, during sequence training as described above, only CS1 was used, CS2 was never used for sequence training, neither for training left eyelid CRs at ISIs other than 500 ms nor for right eyelid CR training.

After successful acquisition of ipsi- or contralateral sequence of CRs, subjects were then presented with occasional CS2-alone probe trials. This input should elicit left eyelid CRs, since this is

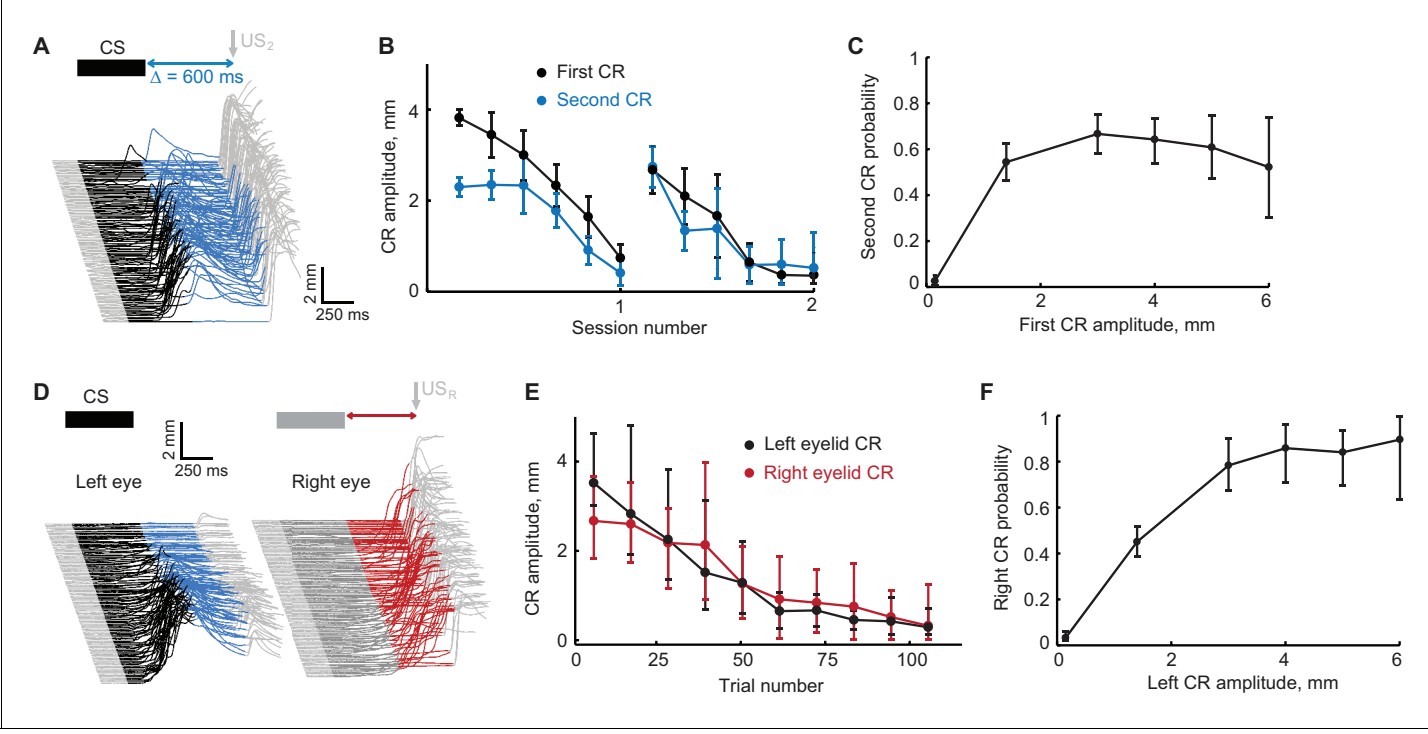

**Figure 4.** Extinction of the first CR in the sequence eliminates the following CRs. (**A**) Schematic representation of stimulus sequence and example session from subject with ipsilateral sequence training. On all paired trials the $US_1$ was never presented and $US_2$ was always presented, regardless of first CR amplitude. (**B**) First and second CR amplitudes over two consecutive sessions of first CR extinction. Each point is an average across one sixth of trials in the session, error-bars represent standard error. (**C**) Second CR probability as a function of first CR amplitude. Error-bars show 95% confidence intervals (obtained by bootstrapping with 0.3 mm non-CR threshold, 2000 repetitions). (**D–F**) Similar analysis for subjects trained in contralateral sequence protocol where black traces represent left eye (first) CRs and red traces represent right eye (second) CRs.

DOI: https://doi.org/10.7554/eLife.37443.010

The following source data and figure supplement are available for figure 4:

**Source data 1.** Behavioral data from the first control experiment: extinction of the first CR while reinforcing the second CR.
DOI: https://doi.org/10.7554/eLife.37443.012

**Figure supplement 1.** Extinction of the first CR in the sequence eliminates the following CRs.
DOI: https://doi.org/10.7554/eLife.37443.011

how subjects were trained. However, the essential test is whether the second CRs in the sequence are also present on CS2-alone trials. Because the $US_2$ ($US_R$) was never presented during the CS2 trials, the presence of second CRs would indicate that the cerebellum did not use CS1 to learn the second CR in the sequence, but rather used FS from the first CR. Again, this is indeed what we observed. Example eyelid responses on CS2-alone trials are shown in *Figure 5B,F* for subjects trained in ipsilateral or contralateral sequence respectively. On most CS2-alone trials with the first left eyelid CR present we also observed the rest of CRs in the sequence the subject was trained to with CS1. The summary across all sessions with CS2 test trials is shown in *Figure 5C,G*. Here each dot shows the probability within a session of the second CR in a sequence, the color indicates a group of trials during which the second CR probability was calculated, based either on CS type (CS1 or CS2) or amplitude of the first left eyelid CR. On trials where the amplitude of first left eyelid CR was larger than 3 mm, the probability of observing other responses in a sequence was the same on trials with either CS1 or CS2 (Tukey's post hoc test, p=0.73, blue versus brown bars for ipsilateral sequence; p=0.51; red versus brown bars for contralateral sequence). Importantly, on CS2 trials without the first response (violet bars), there were no other CRs in the sequence (ipsilateral sequence: one-way ANOVA comparing brown versus violet bars: $F_{(3,61)} = 187$, $p<10^{-28}$; Tukey's post hoc test $p<10^{-9}$; contralateral sequence: one-way ANOVA comparing brown versus violet bars: $F_{(3,62)} = 159$, $p<10^{-29}$; Tukey's post hoc test, $p<10^{-9}$). In addition, we repeated CS2-alone test sessions with

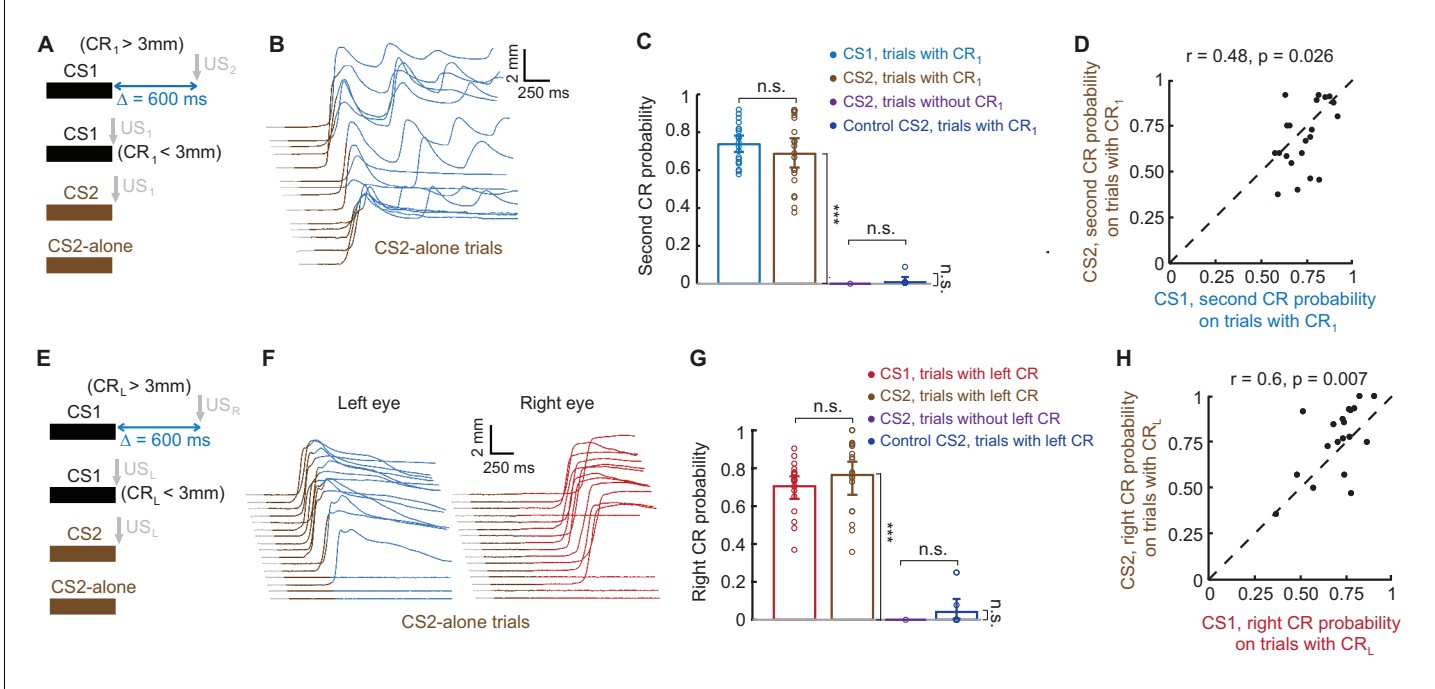

**Figure 5.** Sequence of CRs is present regardless of CS type that drives the first CR. (A) Schematic representation of stimuli during all types of trials in the session. (B) Example session showing eyelid responses on CS2-alone trials of subject trained in ipsilateral sequence with CS1. Brown color indicates CS2 duration. (C) Second CR is present on either CS1 or CS2 trials only if first CR is present. Each dot represents average second CR probability over corresponding trial type during one session, bars show a global average across all CS2 test sessions. Error-bars indicate 95% confidence intervals (bootstrap with 2000 repetitions). (D) Second CR probability on trials with first CRs, elicited either by CS1 or CS2, co-varies between sessions. The data are the same as in blue and brown bars in panel (C). Each dot represents a single session, showing second CR probability on trials with first CRs elicited by either CS1 (X axis) or CS2 (Y axis). (E–H) Similar analysis for subjects trained in contralateral sequence protocol.

DOI: https://doi.org/10.7554/eLife.37443.013

The following source data is available for figure 5:

**Source data 1.** Behavioral data from the second control experiment: CS2 test
DOI: https://doi.org/10.7554/eLife.37443.014

subjects either not trained with (N = 1 and N = 3 for ipsi- and contralateral sequence respectively) or extinguished from (N = 4 and N = 2 for ipsi- and contralateral sequence respectively) producing a sequence of CRs. In this case, while first left eyelid CRs amplitudes were larger than 3 mm, CS2 alone trials did not elicit other CRs in the sequence (ipsilateral sequence: dark blue bars, second CR probability = 0.006 ± 0.008, comparing violet versus dark blue bars with Tukey's post hoc test results in p=0.99; contralateral sequence: dark blue bars, right CR probability = 0.041 ± 0.028, comparing violet versus dark blue bars with Tukey's post hoc test results in p=0.84). In addition, for subjects trained to produce a sequence of CRs, we observed a significant correlation across sessions between probabilities of second CRs in the sequence on CS1 and CS2 trials (*Figure 5D,H*, r = 0.48, p=0.026 for ipsilateral sequence, r = 0.6, p=0.007 for contralateral sequence). Thus, on sessions with a more robust sequence of CRs performed on CS1 trials, subjects also showed larger probability of producing a sequence of CRs on CS2-alone trials. Together, these experiments demonstrate that while the first CR in the sequence is driven by the mossy fiber stimulation CS, CRs that follow are associated with FS from the previous CR.

## Prediction 3: expression of the first CR is necessary for the second CR

Two additional predictions can be tested through additional analyses of the normal sequence-training sessions. Parallel to the logic of the first control experiment described above, we should expect that on trials without a first CR there should be no second CR. To test this prediction we used eyelid responses on CS-alone trials from sessions where the overall probability of second CRs was greater

than 40%. Panels A and B in *Figure 6* show two example trials from subjects trained respectively in ipsilateral and contralateral sequence of CRs. Data from all subjects on all CS-alone trials are shown in *Figure 6C,D*. The amplitude of the second CR plotted versus amplitude of the first CR for subjects trained in ipsilateral sequence of CRs is shown in *Figure 6C*. Each dot represents data from a single CS-alone trial. Similarly, *Figure 6D* shows right eyelid CR amplitudes versus left eyelid CR amplitudes for subjects trained to produce contralateral sequence of CRs. Data corresponding to the example trials are indicated by grey dots. In both plots, horizontal and vertical solid black lines represent nonCR cutoffs (CR amplitudes <0.3 mm), diagonal is shown by a dotted black line. While there were trials with first CR, but no second CR (dots below horizontal line on the bottom), there were no trials without a first CR, but with the second CR (to the left of vertical line). *Figure 6E and F*

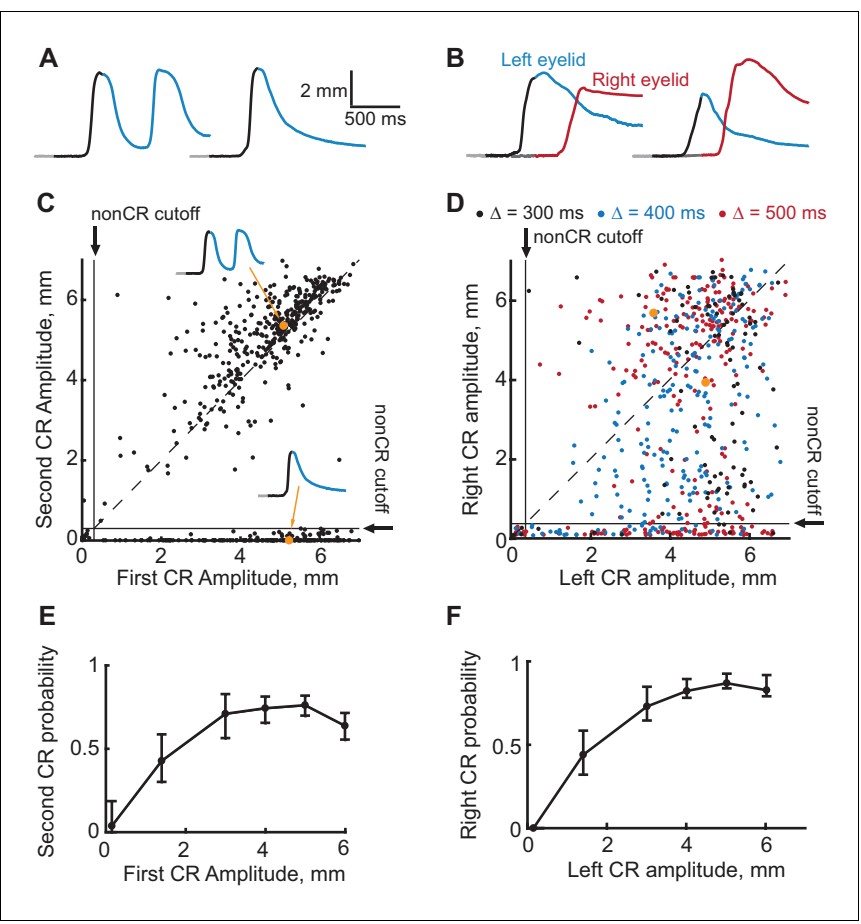

**Figure 6.** Evidence that the expression of a first CR is necessary for expression of subsequent second CR. (**A–B**) Eyelid CRs on two example trials from ipsilateral and contralateral sequence training sessions respectively. Color-coding of time intervals is preserved from *Figures 2* and *3*. (**C**) Scatter plot representing the amplitude of the second CR versus the amplitude of the first CR in ipsilateral sequence training. Each dot represents a single CS-alone trial. Vertical and horizontal solid black lines represent non-CRs cutoffs, dotted black line shows a diagonal where second CR amplitude equals that of the first CR. Dots corresponding to example trials from panels (**A–B**) are shown in orange. (**D**) Similar to (**C**), but for subjects trained in contralateral sequence protocol. Data obtained from sessions with different gap intervals are color-coded as indicated in legend. (**E**) Average probability of the second CR as a function of first CR amplitude. Error-bars show 95% confidence intervals (obtained by bootstrapping with 0.3 mm nonCR threshold, 2000 repetitions). (**F**) Same as (**E**), but for subjects trained at contralateral sequence of CRs. Here we combined data from sessions with different gap intervals.
DOI: https://doi.org/10.7554/eLife.37443.015

The following source data is available for figure 6:

**Source data 1.** Behavioral data on CS-alone trials from sessions with robust expression of CR sequences.
DOI: https://doi.org/10.7554/eLife.37443.016

show the same data plotted as a probability of the second CR (or right eyelid CR), conditioned on the first CR amplitude. For both protocols, the probability of the second CR started to decrease on trials with the first CR amplitude smaller than 3 mm, which was used during sequence training (Chi-square analysis, $\chi^2(6, 571)=70.13$, $p<10^{-13}$, $\chi^2(6, 578)=92.4$, $p<10^{-20}$, for ipsilateral and contralateral sequence protocols respectively). Most importantly, on trials without a first CR, the probability of second CRs was negligible (probability of second left eyelid CR = $0.037 \pm 0.047$ for ipsilateral sequence protocol, probability of right eyelid CR = 0 for contralateral sequence protocol).

## Prediction 4: timing of CRs in a sequence should co-vary on a trial-by-trial basis

There is a natural trial to trial variability in the timing of the CRs. If a FS from the first CR is a signal for the second CR, then it follows that the timing of the first and second CRs should co-vary on a trial by trial basis. For example, on trials with relatively early first CRs the following CRs should also happen earlier than average; on trials with late first CRs the following CRs should also happen later. Example trials with earlier (on top) and later (on bottom) first CR onset times are shown in *Figure 7A and B* from ipsilateral and contralateral sequence respectively. One can notice even from example trials that the whole sequence of CRs is shifted with the timing of the first CR. We investigated the degree of co-variation between timing of CRs in sequence using a variety of CR timing measures, spanning from CR onset time to CR peak time. These timing measures are indicated by orange dots on eyelid position profile in *Figure 7A*. The timing measures were defined for every trial when the amplitude of both responses in a sequence was larger than 2 mm (except for analysis of third and later CRs in ipsilateral sequence, only CS-alone trials were used there). Data from two of pairs of timing measures are shown for each protocol in *Figure 7C–F*. Here each dot represents a single trial, with color indicating the pair of CRs for ipsilateral sequence or the gap interval for contralateral sequence. We applied two types of analyses to investigate the temporal dependence of CRs in the sequence. First, for every pairwise combination of CR timing measures we calculated a Pearson correlation coefficient. We found that the majority of combinations of measures showed a significant trial-to-trial co-variation (*Figure 7—figure supplement 1*), for both ipsilateral and contralateral sequence of CRs. Second, for every trial of each CR timing measures we calculated an inter-CR time interval; correspondingly color-coded distributions of inter-CR time intervals are shown on the right of each panel in *Figure 7C–F*. We than randomly permuted the timing of the first CR and calculated a new shuffled distributions of inter-CR intervals shown in grey. If the timing of the second CR is independent of the timing of the first CR, the shuffled distributions should be identical to the true distributions. We found however a significant difference in distributions for timing measure pairs shown in *Figure 7* (Kolmogorov-Smirnov test, see *Figure 7—source data 1*). These results show that the timing of the second CR in sequence is influenced by the timing of the first CR and support the hypothesis that a FS from the first response drives the learning and expression of the second response.

## Eyelid Purkinje cells encode similarly all CRs in the sequence

The cerebellar cortex has been shown previously to be necessary for acquisition and expression of well-timed eyelid CRs (*Garcia and Mauk, 1998*; *Kalmbach et al., 2010b*). Purkinje cells (PCs), the principal neurons and sole output of the cerebellar cortex, have been a target of several studies. Because PCs are inhibitory neurons, decreases in their activity result in increased cerebellar output that drives eyelid CR (*Heiney et al., 2014*). For single-component eyelid CRs, the timing and magnitude of the decrease in PCs activity have been shown to encode the kinematic features of CRs (*Halverson et al., 2015*; *ten Brinke et al., 2015*). Two studies have also shown that under training conditions that produce two CRs, eyelid PCs also showed two correspondingly timed decreases in activity (*Halverson et al., 2015*; *Jirenhed et al., 2017*). Unlike the present study, in these previous studies both CRs were elicited by the external CS. Thus, to begin to understand the cerebellar mechanisms of movement sequences involving feedback signals, we investigated PC activity during the ipsilateral sequence protocol. If the second CRs are mediated by the usual cerebellar mechanisms, simply driven by a FS rather than CS, then: (1) the same PCs should control kinematics of both first and second CRs, since both responses are produced by the same muscle, and (2) the relationship

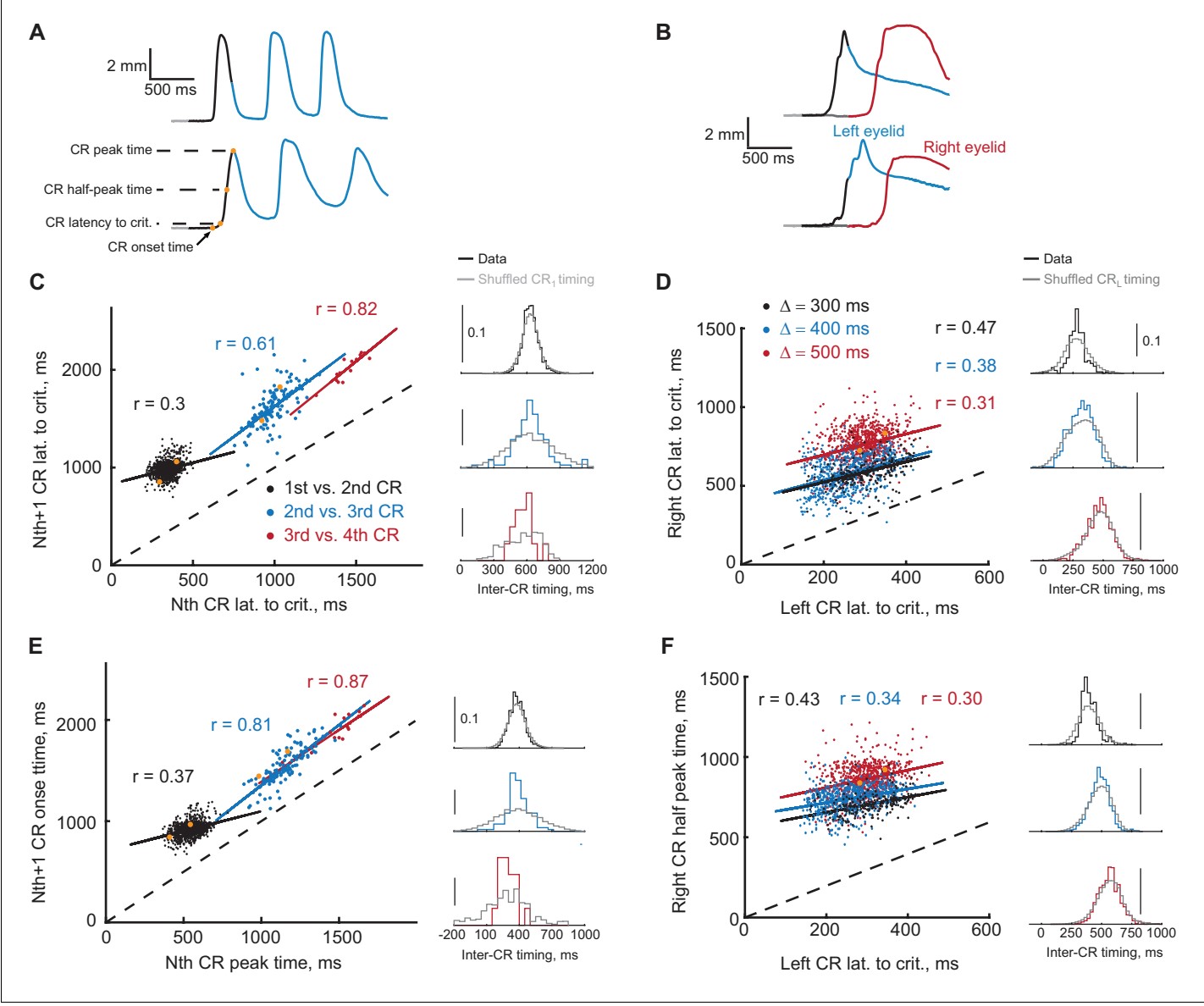

**Figure 7.** Timing of first and second CRs in sequence co-varies from trial-to-trial. (A) Two example trials showing eyelid CRs from subjects trained in ipsilateral sequence. Orange dots indicate times of different CR timing measures, specified on the left. (B) Similarly, responses on two example trials from subjects trained in contralateral sequence. (C) Latency to criterion of following versus previous CR in ipsilateral sequence training. Each dot represents a single trial, dotted black line shows the diagonal. Colored lines show a linear regression fit for corresponding pairs of CRs as in the legend. Colored distributions on the right show distribution of time-intervals between timing measures (CR lat. to crit. versus CR lat. to crit. for C) of corresponding CRs pairs on different trials. Distributions in grey show the same data with timing of first CR shuffled across trials (2000 repetitions). (D) Similar plot for latency to criterion of CRs in contralateral sequence. Here colors indicate different gap intervals, as in the legend. Panels on the right show distributions of timing between left and right CRs data on the left. (E) Same for CR onset time versus peak time of the previous CR in ipsilateral sequence. (F) Same for the time to half peak amplitude of right CR versus latency to criterion of the left CR in contralateral sequence. For panels (C–F) dots corresponding to example trials from panels (A–B) are shown in orange.

DOI: https://doi.org/10.7554/eLife.37443.017

The following source data and figure supplement are available for figure 7:

**Source data 1.** Statistical results of comparison between actual and shuffled distributions of inter-CR timing.
DOI: https://doi.org/10.7554/eLife.37443.019

**Figure supplement 1.** Co-variation of timing measures of CRs in sequence.
DOI: https://doi.org/10.7554/eLife.37443.018

between PC activity and both responses timing and kinematics should be similar. We tested these predictions with in vivo recordings and analysis described below.

Tetrode microdrives were chronically implanted in three subjects targeting the region of cerebellar cortex previously shown to be necessary for acquisition and expression of well-timed eyelid CRs (*Halverson et al., 2015*; *Garcia and Mauk, 1998*; *Kalmbach et al., 2010b*). These subjects were trained to produce the ipsilateral CRs sequence, as before employing electrical stimulation of mossy fibers as the CS. We recorded 156 well-isolated single units during ipsilateral CRs sequence training sessions. Out of those, 42 were classified as PCs based on the presence of both simple and complex spikes and out of those 16 were classified as eyelid PCs (*Halverson et al., 2015*) based on the US-evoked complex spikes. All subsequent analyses involved only these eyelid PCs. Data from an example recording session are shown in *Figure 8A and B*, with eyelid CRs at the bottom and the raster plot showing eyelid PC simple spikes with corresponding PSTH at the top. Panel A shows trials with first CR amplitude smaller than 3 mm, resulting in delivery of $US_1$. Simple spike activity of eyelid PCs developed a decrease during first CR expression, consistent with previous studies. Panel B shows eyelid PC activity and behavioral responses on trials with first CR amplitude larger than 3 mm, resulting in omission of $US_1$ and delivery of $US_2$. Even in this single example it is readily apparent that the same PC develops a decrease in activity corresponding in time to each of CRs in ipsilateral sequence.

Previous work has shown that for a single-component CR, the timing of behavioral CRs and the timing of decreases in eyelid PCs activity co-vary (*Halverson et al., 2015*). We therefore investigated if the same relationship holds for both first and second CRs in ipsilateral sequence protocol. First, as a replication of previous findings, we studied the timing of eyelid PCs responses as a function of first CRs timing. For that, we separated trials into three groups based on first CRs onset times (non-CRs, early CRs and late CRs) and calculated corresponding average firing rates of eyelid PCs. Results related to the first CR timing are shown in *Figure 8C*. The timing of decrease in eyelid PCs firing rate corresponded to the timing of first CRs, similar and consistent with published results (*Halverson et al., 2015*; *ten Brinke et al., 2015*). Namely, on non-CR trials (black lines) eyelid PCs firing rate barely deviated from baseline activity. On trials with early first CRs (red lines) decreases in eyelid PCs firing rate happened robustly earlier than on late first CR trials (blue lines). For results shown here and below, the absence of overlap in 95% confidence intervals between eyelid PCs activity corresponding to different groups was used as evidence of reliable separation. Next, we investigated whether the same relationship holds for the timing of second CRs in the sequence and eyelid PC activity. Results demonstrating that the timing of the second decrease in eyelid PCs activity also matches the timing of second CRs are shown in *Figure 8D*. Here we similarly separated trials into three groups, now based on the presence and timing of the second CRs. Now on trials without the second CRs (black lines), PCs firing rate returned to the baseline level after the decrease corresponding to the first CR. On trials with second CRs present (red and blue lines) PCs activity demonstrated double decrease corresponding to the first CR and second CRs in the sequence. Moreover, the timing of the second decrease in PCs activity followed the timing of second CRs (*Figure 8D*, early versus late second CRs shown by red and blue lines correspondingly).

Previous work (*Halverson et al., 2015*) demonstrated that for single-component CRs, the decrease in eyelid PCs activity precedes the onset of CR, consistent with the causal relationship between eyelid PCs responses and CRs expression. To examine whether changes in eyelid PCs activity precede the expression of first and second CRs in the ipsilateral sequence, we aligned eyelid PCs activity to the onset of first and second CRs (*Figure 8E and F*). *Figure 8E* shows first CR time profile (top) and corresponding PCs activity (bottom) aligned to the first CR onset time (black vertical dotted line). Purkinje cells activity was calculated on first CR trials (blue line) and non-CR trials (black line). Non-CR trials were aligned by CR onset by randomly sampling from the distribution of first CR onset times. Similar to published results during conventional eyelid conditioning sessions (*Halverson et al., 2015*), PCs activity on CR trials demonstrated a robust decrease in activity prior to CR onset. We next performed a similar analysis for the second CRs, with results shown in *Figure 8F*. Eyelid PCs activity reliably separated prior to the second CR onset on trials with second CRs (blue line) comparing to non-second-CRs trials (black line). The amount of decrease in PCs activity from baseline at the moment of CR onset was similar for first and second CRs ($0.766 \pm 0.015$ at the time of first CR onset; $0.796 \pm 0.034$ at the time of second CR onset; p=0.45, Wilcoxon rank sum test).

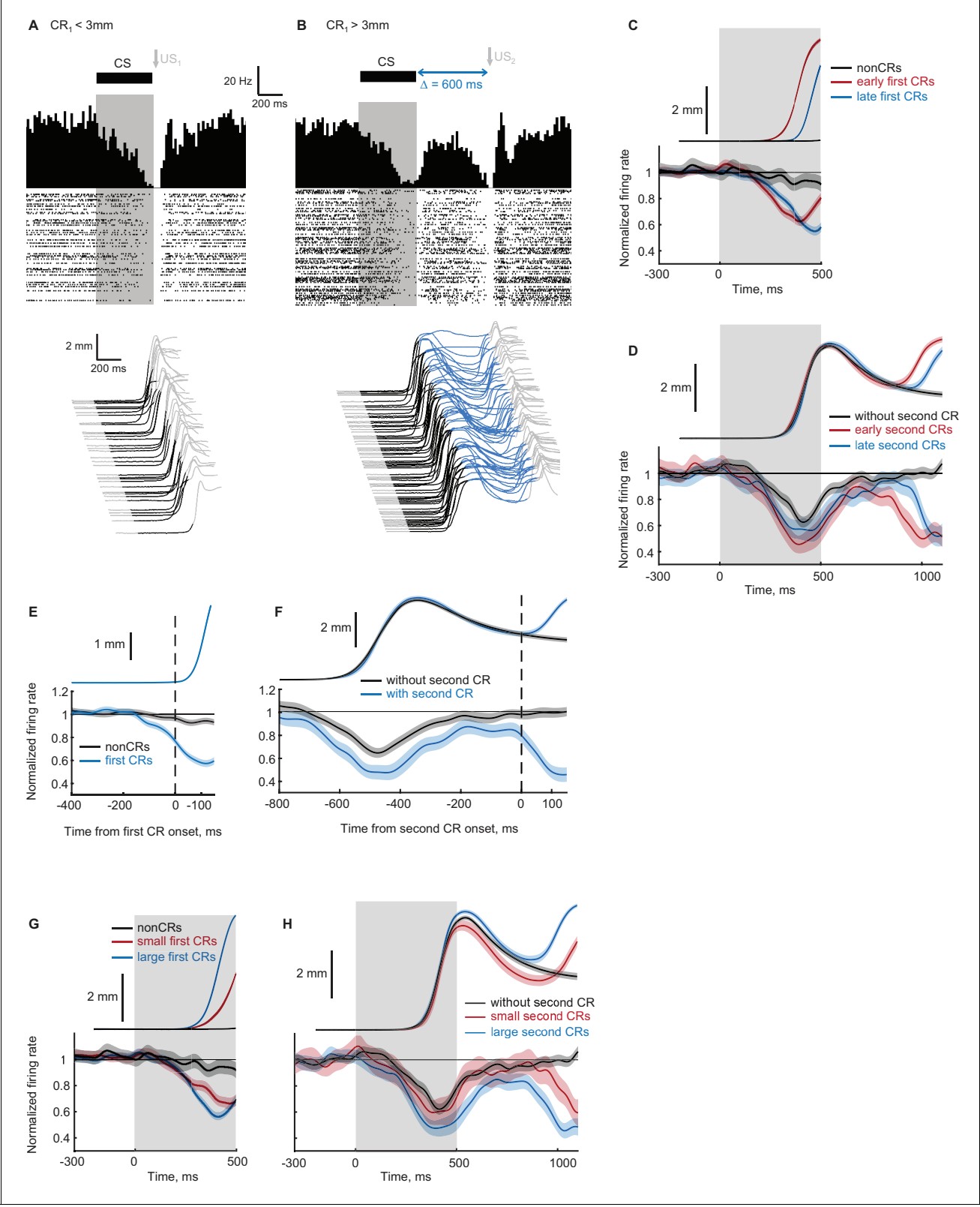

**Figure 8.** Recordings from eyelid Purkinje cells during ipsilateral sequence sessions. (A–B) Example eyelid PC recording during an ipsilateral sequence training session. (A) Behavioral responses on trials with first CR amplitude smaller than 3 mm are shown at the bottom, corresponding PSTH and a raster plot of eyelid PC simple spikes are shown at the top. CS duration is indicated by a black color in behavioral waterfall plots and by a grey shaded area in the raster plot. (B) The same format as in panel (A), showing data on trials with first CR amplitude larger than 3 mm and consequently $US_2$
*Figure 8 continued on next page*

*Figure 8 continued*

delivery. (C) Average eyelid response profiles on trials sorted by the onset time of first CR are shown on top, corresponding average eyelid PCs firing rate normalized by the baseline level is shown at the bottom. Non-CR trials are shown in black, trials with early and late CR onset times are shown in red and blue respectively. Behavioral responses and eyelid PCs activity are truncated at $US_1$ onset, shaded regions represent 95% confidence intervals. (D) Similar to (C), but for trials with first CR present and sorted by the onset times of second CRs. Color-coding is similar to (C), behavioral responses and eyelid PCs activity are truncated at $US_2$ onset. (E) Eyelid PCs activity aligned by first CR onset times (vertical black dotted line) is shown at the bottom; aligned CRs profile is shown on top. Eyelid PCs activity on CR trials is shown in blue color, on non-CR trials – in black. (F) Behavioral responses and eyelid PCs activity aligned to the onset time of second CR (vertical black dotted line). First CRs are present in all trials, results from trials with second CRs present are shown in blue color, without second CRs – in black. (G) Average eyelid response profiles on trials sorted by first CR amplitude are shown on top, corresponding average eyelid PCs firing rate is shown at the bottom. Non-CR trials are shown in black, trials with small and large first CR amplitudes are shown in red and blue respectively. (H) Similar to (G), but for trials with first CR present and sorted by the second CR amplitude.

DOI: https://doi.org/10.7554/eLife.37443.020

The following source data is available for figure 8:

**Source data 1.** Normalized eyelid Purkinje cell firing rate and corresponding behavioral data during expression of ipsilateral sequence of CRs.

DOI: https://doi.org/10.7554/eLife.37443.021

The above results demonstrate that individual eyelid PCs show decreases in activity that precede the onset and match the presence and timing of both first and second eyelid CRs in an ipsilateral sequence. We next examined whether this similarity holds for the relationship between eyelid CRs amplitude and the amount of PCs decrease, as has been demonstrated for the single-component CRs (*Halverson et al., 2015*; *ten Brinke et al., 2015*). First, as additional replication of previous findings, we separated trials into three groups based on the first CR amplitude and calculated average eyelid PCs responses (*Figure 8F*). As expected, the amount of decrease in eyelid PCs activity scaled with first CR amplitude. We next investigated if this relationship holds for the second CRs. For that we performed a similar analysis by separating trials based on the second CR amplitude (*Figure 8G*). Eyelid PCs activity showed no change on trials without the second CR (black line) and correspondingly scaled amount of decrease on trials with smaller and larger second CRs (red and blue lines respectively). Overall, recordings from eyelid PCs demonstrate that individual PCs learn both first and second components in a sequence and the way PC responses relate to these components is not distinguishable. This suggests that the same cerebellar mechanisms are involved in generation of both CRs in the sequence. These data are consistent with the general notion that FS can be used by the cerebellum to learn following responses in the sequence through processes similar to learning the first response using the CS.

## Discussion

We investigated how movement sequences can be learned and expressed by the cerebellum. The data demonstrate that feedback signals from a CR are a sufficient signal for the cerebellum to learn the next response in a sequence (*Figure 1C*). This finding held true for both ipsilateral and contralateral sequences of eyelid CRs. With a series of control experiments we showed that responses following the first CR are associated with feedback signals from the first CR and not with the electrical stimulation of mossy fibers that signaled the first response in the sequence. Finally, recordings from eyelid Purkinje cells, the sole output neurons of cerebellar cortex, show that all CRs in the sequence are encoded in the same manner by the cerebellum and hence share similar neural learning mechanisms. These results demonstrate how, through simple associative learning processes, the cerebellum can learn to chain together a sequence of appropriately timed movements using mossy fiber inputs that include feedback signals from previous movement components. These results also show that, in principle, a cerebellar-mediated movement sequence can be initiated by a discrete mossy fiber signal present only at the beginning of the sequence. Feedback signals can then, to a point, sustain subsequent components.

### Possible sources of feedback signals

While we have demonstrated the sufficiency of FS from an eyelid CR to be used by the cerebellum as a new 'CS', the nature and source of this FS remains an open question. Several sources of feedback are possible. The most direct candidate feedback route would originate from the deep

cerebellar nucleus neurons (DCN), which provide the output of the cerebellum. Increases in DCN neuron activity are known to drive the expression of eyelid CRs (*Halverson et al., 2010*; *McCormick and Thompson, 1984*). DCN neurons are also known to have axon collaterals that return to the cerebellar cortex (*Houck and Person, 2015*; *Ankri et al., 2015*; *Gao et al., 2016*) and form mossy-fiber like synaptic connections. Thus the information about the ongoing expression of a CR can be passed back to granule cells in the cerebellar cortex via a monosynaptic excitation by axon collaterals of glutamatergic DCN neurons (*Houck and Person, 2015*; *Gao et al., 2016*) or via a reduction of inhibition from Golgi cells that receive inhibitory projections (*Ankri et al., 2015*) from GABA/glycinergic DCN neurons. More indirect routes to convey DCN activity back to cerebellar cortex also exist, for example via thalamic (*Halverson et al., 2010*) or red nucleus feedback (*Cartford et al., 1997*) to pontine nuclei neurons. It is also possible that the FS may not originate directly from cerebellar output activity, but rather from proprioceptive information driven by the movement itself. All of these possible routes can contribute to a FS, none are mutually exclusive, and their relative contributions will likely depend on temporal constrains and/or laterality of movement sequence. For example, nucleo-cortical feedback routes have been shown (*Houck and Person, 2015*) to have more dominant ipsilateral projections to areas of cerebellar cortex. This can be a possible explanation for the need to use shorter time-intervals for subjects trained in contralateral sequence comparing to ipsilateral. While we know that the feedback information is supplied to the cerebellar cortex (*Giovannucci et al., 2017*), further studies are needed to investigate relative contributions of possible feedback pathways.

Apart from the cerebellum, the basal ganglia is another prominent brain system implicated in learning and execution of movement sequences (*Doyon et al., 1997*; *Lehéricy et al., 2005*; *Seidler et al., 2005*). However the relative involvement of the cerebellum and basal ganglia in learning and production of motor sequences might differ. While several imaging (*Lehéricy et al., 2005*; *Doyon et al., 2002*; *Seidler et al., 2005*) and recording (*Jog et al., 1999*) studies have reported changes in basal ganglia activity during sequence learning, studies involving lesions or neurodegenerative diseases were generally less conclusive. Studies in Parkinson's patients showed only a partial (*Doyon et al., 1997*; *Shin and Ivry, 2003*; *Jackson et al., 1995*) impairment of sequence learning compared to control groups and patients with focal basal ganglia lesions did not display deficits (*Shin et al., 2005*). Moreover, pharmacological lesion of globus pallidus internus, the primary basal ganglia output region, resulted in decrease of movement velocity and acceleration, but did not impair the production of learned sequences of movements (*Desmurget and Turner, 2010*). While we addressed in our work the cerebellar involvement in learning a movement sequence, the results do not preclude significant basal ganglia contributions to learning and expression of movement sequences.

## Cerebellar contribution to complex sequences

In our experiments we trained subjects to produce a sequence of discrete responses. The results of our study however should not rely on this aspect of the responses, which we utilized for the clarity of analysis. Natural complex multi-joint movements, such as locomotion or reaching movements, can be broken down into a sequence of several movement components. In these cases our results imply that the feedback about the each portion of the movement can be used by the cerebellum to learn a correct continuation of the movement.

This general framework can also be used to describe how the cerebellum can learn to guide cortical activity to produce a desired sequence of states. The cerebellum and cerebral cortex areas, such as premotor and primary motor cortices, are known to form a closed loop system via cortico-ponto-cerebellar projections (*Kelly and Strick, 2003*; *Glickstein et al., 1985*; *Evarts and Thach, 1969*), where the cerebellum sends information back to the cerebral cortex via the thalamus. Electrical (*Penfield and Boldrey, 1937*; *Graziano et al., 2002*) or optogenetic (*Harrison et al., 2012*) stimulation of motor cortex area is known to produce complex multi-joint movements. A common interpretation of these results is a notion of motor program (*Mink, 1996*; *Summers and Anson, 2009*), stored either entirely within a motor cortex or at least partially within downstream areas. Our study provides new evidence towards an interpretation that motor programs could be stored in part within the cerebellum. This way, the initial command from motor cortex serves as a trigger to initiate the movement, while the later feedback signals from cerebellar output and movement itself are used by the cerebellum to learn a proper output that guides motor cortex activity along the correct

trajectory in state space, which in turn results in a correct movement trajectory. While speculative at this point, such a framework is fully compatible with results of the present study.

## Materials and methods

### Surgery

In all experiments subjects were 14 male New Zealand albino rabbits weighing 2.5–3.5 kg at experiment onset. Treatment of rabbits and surgical procedures were in accordance with National Institutes of Health guidelines and an institutionally approved animal welfare protocol. All subjects were maintained on a 12 hr light/dark cycle. One week before the start of experiment, subjects were removed from the home cage and anesthetized with a cocktail of acepromazine (1.5 mg/kg) and ketamine (45 mg/kg). After onset of anesthesia, the subjects were placed in a stereotaxic frame, intubated, and maintained on isoflurane (1~2% mixed in oxygen) for the remainder of the surgery. Under sterile conditions the skull was exposed with a midline incision (~5 cm), and four holes were drilled for anchor screws. Some anchor screws also functioned as ground screws for subjects with mossy fiber stimulation implants or a microdrive ground for subjects with microdrive implant. The rabbit's head was then positioned with lambda 1.5 mm ventral to bregma.

For subjects prepared only for behavioral experiments involving electrical stimulation of mossy fibers, a craniotomy was drilled out at 5.5 mm lateral and 3 mm anterior from lambda, ipsilateral to the trained eye. Skull fragments were carefully removed from the craniotomies, the dura matter was carefully opened under visual guidance. One or two laterally spaced (by 1 mm) tungsten stimulating electrodes (A-M Systems, Carlsborg, WA; tip exposed to obtain impedance of 100–200 kΩ) were implanted in the middle cerebellar peduncle (16 mm ventral to lambda).

For subjects prepared for microdrive implantation in the cerebellar cortex, a craniotomy was also drilled out at 5.9 mm posterior and 6.0 mm lateral to lambda. Skull fragments were carefully removed from the craniotomies, the dura matter was carefully opened under visual guidance. A custom-made microdrive (16 tetrodes and 2 references) fitted with an electronic interface board (EIB-36-16TT, Neuralynx) was implanted in the left anterior lobe of the cerebellar cortex at a 40° angle posterior to vertical and 17.8 mm ventral to lambda. This region of the anterior lobe has been shown to be involved in acquisition and expression of well-timed conditioned eyelid responses (*Halverson et al., 2015*; *Kalmbach et al., 2010a*; *Garcia et al., 1999*). The primary target of tetrode recordings were PCs with evoked complex spikes from US, referred throughout the manuscript as eyelid PCs. The bundle cannula of the microdrive was lowered to the surface of the brain. The craniotomies were sealed with low viscosity silicon (Kwik-Sil; World Precision Instruments). The head bolt to mount eyelid detector, anchor/ground screws, stimulation electrodes and microdrive were secured with dental acrylic (Bosworth Fastray, Pink; The Harry J. Bosworth Company), and the skin was sutured where the skull and muscle were exposed. Finally, two stainless steel loops terminating in gold pins were inserted into the anterior and posterior periorbital region of the left eye (and optionally right eye for subjects trained in contralateral sequence of CRs) for delivery of the stimulation US. Subjects were given postoperative analgesics and antibiotics for 2 days after surgery and were allowed to recover for a week before experiments began.

### Conditioning

The subjects were trained in custom-designed, well-ventilated, and sound attenuating chambers measuring 90 × 60 × 60 cm (length, width, height). Each rabbit was placed in a plastic restrainer with their ears stretched over a foam pad and taped down to limit head movement. To measure eyelid position, an infrared emitter/detector system was attached directly to the head stage of each rabbit to record movements of the left external eyelid. These detectors provide a linear readout of eyelid position (±0.1 mm) at 1 kHz sampling rate by measuring the amount of infrared light reflected back to the detector, which increases as the eye closes (*Ryan et al., 2006*). At the start of each daily session, the gain of the eyelid position detector was calibrated by delivering a test US to elicit full eyelid closure (defined as 6.0 mm, typical for an adult rabbit). A trial would not start until the rabbit's eyelid was sufficiently open. Stimulus presentation was controlled by custom-designed software for all experiments.

## Stimulus delivery

Each conditioning chamber was equipped with a speaker that was connected to a stereo equalizer and receiver which were connected to a computer that generated the tone. For subjects trained using tone as the CS, the CS was set as a 1 kHz, 500 ms, 75 dB sinusoidal tone with a rise and fall time of 5 ms to avoid audible clicks from the speaker. For subjects trained with electrical stimulation of mossy fibers, the CS was a constant frequency pulse train of cathodal current pulses (100 Hz, 500 ms, 0.1 ms pulse width, 100–150 µA), generated by a stimulus isolator (model 2300, A-M Systems, Carlsborg, WA) and passed through the electrode(s) implanted in the middle cerebellar peduncle. The current pulses were controlled by custom written software (available from authors upon request) and delivered through an isolated Pulse Stimulator (model 2100, A-M Systems, Carlsborg, WA). Electrical leads from a separate stimulator (model 2100) were attached to the periorbital electrodes to deliver pulses of electrical stimulation to the left eyelid as the US. The US was a 50 ms train of constant current pulses (50 Hz, 0.7–1 ms pulse width, 1–3 mA) delivered through the periorbital electrodes. US intensity was adjusted for each rabbit to produce a full eyelid closure without any pain reactions. All types of trials in all sessions were separated by a mean inter-trial interval of $30 \pm 10$ s.

## Initial training

For initial training, subjects were given daily eyelid conditioning sessions comprised of 12 blocks of 9 trials each. All subjects were initially trained at an inter-stimulus interval (ISI) of 500 ms to produce left eye CRs. Each block consisted of 1 CS-alone trial and 8 paired trials. After reaching a robust CR performance, subjects were switched to the sequence training protocols.

## Sequence training protocols

### Ipsilateral sequence of CRs

Training sessions were comprised of 12 (or 8 in rare cases) blocks of 9 trials each. Each block consisted of 1 CS-alone trial and 8 paired trials. On paired training trials subjects were presented with the same length of mossy fiber stimulation CS (500 ms) used during initial training, but with the US presented at one of two different times: the first time, designated as $US_1$, was at CS offset as with normal training. The second time, designated as $US_2$, occurred at 600 ms after CS offset. The factor that determined whether the US was presented at $US_1$ versus $US_2$ was the amplitude of the CR elicited by the mossy fiber stimulation CS. On trials when CR amplitude was lower than the target (3 mm, corresponding to half-sized CR), $US_1$ was presented. The purpose of $US_1$ was to maintain robust responding of the first CR. If amplitude of the first CR was higher than the target, $US_1$ was omitted and $US_2$ was presented. Since $US_1$ and $US_2$ were both delivered to the left eye, their intensity was the same. Some parameters of US (pulse width and intensity) were adjusted per subject to decrease the amount of squinting the eyelid while still being sufficiently strong to support learning and expressions of CRs.

### Contralateral sequence of CRs

Training sessions were comprised of 12 (or 8 in rare cases) blocks of 9 trials each. Each block consisted of 1 CS-alone trial and 8 paired trials. On paired training trials subjects were presented with the same length of mossy fiber stimulation CS (500 ms) used during initial training, US delivery was automatically determined by the following rule: if left eye CR amplitude was lower than the target (3 mm, half-sized CR), $US_L$ was presented to the left eye to maintain robust responding of the left eye CR. If amplitude of the left CR was higher than the target, $US_L$ was omitted and $US_R$ was presented to the right eye. During initial acquisition, the interval between CS offset and $US_R$ was typically 400 ms (N = 4), but for some subjects was 300 ms (N = 1) or 500 ms (N = 1). We chose to use a shorter duration of the gap interval compared to ipsilateral sequence training, because our pilot data showed that most subjects were unable to learn a contralateral sequence with 600 ms gap interval from naïve right eye state. After acquisition of contralateral sequence of CRs subjects were switched to 500 ms gap interval training in case shorter gap duration was used initially.

### Extinction of following responses in the sequence

The modification to session design from initial training was straightforward. First left eye CRs elicited by mossy fiber stimulation CS were still reinforced with $US_1$ ($US_L$) if CR amplitude was smaller than 3

mm. However on trials with left eye CR larger than 3 mm $US_2$ ($US_R$) was no longer delivered compared to the training session.

## Extinction of the first CR in the sequence while reinforcing the following CR

Results from sessions described here are shown in *Figure 4*. Here on paired trials we stopped delivering $US_1$ ($US_L$) regardless of the first left eye CR amplitude, while always delivering $US_2$ ($US_R$). The purpose of $US_1$ ($US_L$) was to reinforce first CR performance, hence its absence lead to extinction of first left eye CRs. If the following responses in the sequence were driven by mossy fiber stimulation CS, they should still maintain because of reinforcing $US_2$ ($US_R$) delivery. Since with such protocol extinction of first left eye CRs was faster in contralateral than ipsilateral protocol, subjects in ipsilateral sequence protocol were given at least two consecutive days of such sessions while for subjects trained in contralateral protocol one session was sufficient in most cases.

## CS2 test sessions

Results from sessions described here are shown in *Figure 5*. Subjects were trained in sequence of CRs protocols using mossy fiber stimulation CS, referred in this section as CS1. In addition, subjects were trained at ISI 500 ms to produce left eye CRs with CS2. CS2 was either also a 500 ms long electrical stimulation of mossy fibers delivered through a separate electrode (N = 3 and N = 2 for subjects trained in ipsilateral or contralateral sequence of CRs) or a 500 ms 1 kHz tone (N = 2 and N = 3 respectively). After robust responding to CS2 was reached, subject were switched to CS2 test sessions. CS2 was never used for sequence training, nor for prior training of left eye CRs at ISIs other than 500 ms, nor was it used for right eye CR training. Each session consisted of 108 trials. Here on 60% of trials CS1 was delivered and sequence of CRs was reinforced with $US_1$($US_L$) or $US_2$($US_R$) as described in training protocol. Another 25% of trials were ISI500 CS2 paired trials. The rest – 15% of trials were CS2-alone trials. We chose these ratios (and slightly adjusted them per subject if needed) to maintain a balance between: high enough number of CS1 trials so that the performance of CRs in the sequence was sufficient, high enough number of paired CS2 trials so that the performance of CRs elicited by CS2 was sufficient and finally high enough number of CS2-alone trials so that the amount of CS2 test trials per session was sufficient.

## In vivo recordings and unit isolation

The details about recording procedure, single units isolation and identification of eyelid Purkinje cells have been published previously (*Halverson et al., 2015*). Briefly, each independently movable tetrode in a microdrive was comprised of four nichrome wires (12 µm diameter; Kanthal Palm Coast). These were twisted and then heated so that the insulation was partially melted together to form a tetrode. The individual wires of each tetrode were connected to the EIB with gold pins. Each tetrode was gold plated to reduce final impedance to 0.5–1 MΩ measured at 1 kHz (nanoZ kit; Neuralynx). During surgery, the tetrodes were placed over the target site of the left anterior lobe of the cerebellar cortex and were advanced to within 2.0 mm ventrally from the target during surgery using stereotaxic guidance. After recovery from surgery each tetrode was lowered in 40–80 µm increments per day until at least one stable single unit was identified, although there were often multiple units on a single tetrode. Typically tetrodes were allowed to stabilize for 24 hr and units were checked again the following day, although on a small fraction of sessions the recording was initiated on the same day if new units appeared to be stable. A custom-written cluster cutting program was used to isolate single units offline. Commonly used waveform features, such as peak, valley and energy were used during cluster-cutting. Additional features, such as the late peak measure (*Halverson et al., 2015*), were used to identify complex spikes and differentiate them from simple spikes. In some cases complex spikes formed a separate cluster from simple spikes when viewed in the peak, valley or energy planes. Belonging of putative simple and complex spikes to the same Purkinje cell was verified by computing a spike-triggered average of simple spikes on complex spikes, demonstrating a post-complex spike pause. Eyelid Purkinje cells were defined by the presence of US elicited complex spikes. The remaining PCs were considered 'non-eyelid'.

## Data analysis
Following cluster cutting, all subsequent analysis of single unit and behavioral data were performed using custom-written scripts in MATLAB (Mathworks).

## Eyelid data analysis
For each trial, 2500 ms of eyelid position data (200 ms pre-CS, 2300 ms post CS) were collected at 1 kHz sampling rate and at 12 bit resolution. The data were stored to a computer disk for subsequent off-line analysis. Eyelid position data were passed through a low-pass Savitzky–Golay filter. Eyelid velocity was calculated as a derivative of eyelid position with a second-order accurate scheme, again passed through a low-pass Savitzky–Golay filter.

A small fraction of trials were discarded if: (1) upward eyelid movement exceeding 0.3 mm CR criterion during 200 ms before the CS onset or first 100 ms from CS onset or (2) negative eyelid deviation below 0.5 mm at any time prior to US onset on paired trials and from the trial start to 500 ms after CS offset on CS-alone trials. As a result, 1.4% (242/17574) of trials from sessions related to ipsilateral sequence protocol and 5.3% (690/13109) of trials from sessions related to contralateral sequence protocol were discarded. A larger fraction of trials discarded in contralateral sequence of CRs sessions was due to the fact that fluctuations of both left and right eyelid positions contributed to the exclusion criteria.

An eyelid response was counted as a conditioned response (CR) if CR amplitude reached the 0.3 mm criterion. The CR onset time was defined only for CR trials and was determined using a custom-written two-step algorithm. The first step was designed to detect the initial deflection of eyelid position away from the pre-CS baseline, while the second step used linear interpolation to determine the exact time of CR onset. CR latency to criterion was defined as the first time point when eyelid position deviated above CR criterion. Additional CR features, such as time to peak of CR velocity, time to half of peak CR, time to 90% of CR peak and CR peak time were introduced and calculated to study the co-variation of timing measures of CRs in the sequence.

Because of complexity of possible eyelid response profiles compared to conventional eyelid conditioning sessions, CR amplitudes were calculated in the following way.

## Ipsilateral sequence of CRs
First, trials were passed through the criterion design to exclude trials where subject squinted the eyelid after the first CR making it not possible to determine the presence or absence of following CR. We calculated the difference between (1) the maximum of eyelid position value in between 400 to 1100 ms from CS onset and (2) the minimum of eyelid position value in between 800 to 1100 ms from CS onset (as a reminder, $US_2$ was delivered at 1100 ms from CS onset). Such difference should have exceeded at least 1 mm for the trial to be analyzed further. On CS-alone trials, MATLAB function *findpeaks* with *MinPeakProminence* = 0.5 was initially used to estimate the number of CRs on a given trial. For trials with number of CRs equal or larger than 2 and first CR onset time smaller than 500 ms (length of CS), *findpeaks* function output was used to determine the number, peak times and corresponding amplitudes of CRs. If the onset time of first CR was larger than 500 ms, we made an estimate of whether the first CR was delayed or absent. Here first CR amplitude was defined as the maximum value eyelid position between CS onset and 200 ms after CS offset; the second CR amplitude was defined as the maximum value of eyelid position between 200 to 800 ms from CS offset. On paired trials a similar set of procedures was performed. The peak time of the second CR was assumed to be within 100 ms of $US_2$. The onset time of the second (and later) CR was defined as the time of the minimum of eyelid position value between two consecutive CRs.

## Contralateral sequence of CRs
For contralateral sequence protocol, a correction was made to account for possible simultaneous movement of right eyelid during left eye CR. Though such instances were rare, occasionally a subject's right eyelid position would deviate from baseline during left eye CR. If such deviation would reach a CR criterion, such movement could potentially be falsely identified as a right eye CR. In order to reduce the number of false positives, right eye CR amplitude was calculated by subtracting the maximum of right eyelid position during CS (when left eye CR would occur) from the overall maximum of right eyelid position.

*Co-variation of timing measures of CRs in sequence*. To study co-variation of timing between all possible points along CR profile, we performed a following procedure (with results reported in *Figure 7—figure supplement 1*). For every CS-alone trial with both CR amplitudes in sequence larger than 3 mm, we:

1. Extracted CR time-profiles in the sequence;
2. Normalized each CR profile by its peak value, such that eyelid position values during a CR range between zero and one;
3. Starting from the CR onset time and up to CR peak time, we binned CR profile in 0.005 increments of its fraction and found a corresponding time point for each increment. The same procedure was repeated for the second CR in the sequence;

After this process was repeated for all trials, for every combination of fraction of CRs in sequence (e.g. first CR onset times versus second CR one-third of peak times), we found a Pearson correlation coefficient of corresponding timing co-variability. A small fraction of trials with outliers in CR timing (defined as larger than 3 standard deviations from the mean value across all trials) were discarded.

### Single unit data analysis

Spike times of individual PCs were synchronized with recordings of eyelid position and stimuli onset/offset times. Spike times were rounded to the nearest millisecond. Instantaneous firing rate of each PC was estimated on every trial using inverse an inter-spike-interval followed by a two-sided Gaussian kernel with a 20 ms standard deviation window. For every PC the firing rate was normalized by the value of the baseline firing rate during 1500 ms of pre CS activity.

To investigate how simple spike activity of eyelid PCs corresponds to the onset and amplitude of CRs in ipsilateral sequence, we performed a grouping procedure. Trials with CRs from sessions with eyelid PCs recordings were combined and divided into three equal subgroups of trials, according to CR onset or amplitude. The number of trials in non-CR group was not controlled and was typically low since subjects were well-trained. For analysis of eyelid PC activity during the first CR, all trial types ($US_1$-trials, $US_2$-trials, CS-alone trials) were used. For analysis of eyelid PC activity during the second CR, only trials with first CR larger than 3 mm were used ($US_2$-trials and some CS-alone trials). Subgroup behavioral and neural data were then averaged within each group and superimposed. The absence of overlap in 95% confidence intervals between groups of average behavioral responses and eyelid PC single unit activity were used as evidence of reliable separation.

### Statistics

All statistical tests were done in MATLAB and are stated in the text. Data are reported as mean ± SEM or mean ±95% confidence interval if noted. In cases when the mean value and confidence intervals of CR probability needed to be computed from CR amplitude data that were combined across several sessions (*Figures 4* and *6*), the values were obtained by performing bootstrap sampling (2000 repetitions) from corresponding CR amplitude distribution, using 0.3 mm CR threshold that corresponds to 5% of full eyelid closure. Sample sizes for each experiment are stated in the text. For comparison between groups, we performed Wilcoxon signed-rank test (paired), Wilcoxon rank sum test (unpaired) or one-way ANOVA with a following post hoc Tukey's test. Difference in distributions was assessed with Kolmogorov-Smirnov test. All p values are reported exactly, unless $p < 0.001$.

## Acknowledgements

This work was supported by MH 46904 and NS98308 to MDM.

## Additional information

### Funding

| Funder | Grant reference number | Author |
| --- | --- | --- |
| National Institute of Mental Health | MH 46904 | Michael Dean Mauk |

| National Institute of Neurological Disorders and Stroke | NS 98308 | Michael Dean Mauk |
|---|---|---|

The funders had no role in study design, data collection and interpretation, or the decision to submit the work for publication.

## Author contributions

Andrei Khilkevich, Conceptualization, Data curation, Software, Formal analysis, Validation, Investigation, Visualization, Writing—original draft, Project administration, Writing—review and editing; Juan Zambrano, Molly-Marie Richards, Data curation, Validation, Investigation; Michael Dean Mauk, Conceptualization, Supervision, Funding acquisition, Writing—review and editing

## Author ORCIDs

Andrei Khilkevich (iD) https://orcid.org/0000-0002-1876-4928

## Ethics

Animal experimentation: Treatment of animals and surgical procedures were in accordance with National Institutes of Health guidelines and an institutional animal care and use committee (IACUC) protocol (#AUP-2015-00137) of the University of Texas at Austin. Every effort was made to minimize suffering.

## Decision letter and Author response

Decision letter https://doi.org/10.7554/eLife.37443.025
Author response https://doi.org/10.7554/eLife.37443.026

## Additional files

### Supplementary files

• Source code 1. Matlab code for source data.
DOI: https://doi.org/10.7554/eLife.37443.022

• Transparent reporting form
DOI: https://doi.org/10.7554/eLife.37443.023

### Data availability

All data needed to evaluate the conclusions in the paper are present in the paper and/or the Supplementary Figures. Source data are provided for Figures 2-8.

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
