## [Decision Letter]

Thank you for submitting your article "A movement drives a movement. Cerebellar implementation of movement sequences through feedback" for consideration by *eLife*. Your article has been reviewed by three peer reviewers, and the evaluation has been overseen by Indira Raman as the Reviewing Editor and Richard Ivry as the Senior Editor. The following individuals involved in review of your submission have agreed to reveal their identity: Reza Shadmehr (Reviewer #1); John Freeman (Reviewer #2).

The reviewers have discussed the reviews with one another and the Reviewing Editor has drafted this decision to help you prepare a revised submission.

Summary:

This study examines the role of the cerebellum in coordinating sequences of movements by analysis of eyelid conditioning in rabbits. The experiments provide evidence that sequences of associatively learned movements are chained together with the feedback signal from one learned response serving as the trigger or cue for the next response.

Essential revisions:

The reviewers all found the work to be interesting and convincing. The "General Assessments" from the reviewers are all included below for the authors' information and satisfaction. Since they are so favorable, we thought the authors would like to read them in their raw form.

The revisions requested focus on data presentation/statistical analysis and addition of some information that is present within the datasets. They are summarized here, but are expanded upon in the "Major Points" section.

1) Data presentation/statistical analysis. The reviewers indicated that in a couple of places, the data should be presented in terms of conditional probabilities. Please see Major points 1A and 1B below.

2) Complex spikes. Since complex spike data is present within the dataset, the reviewers requested that information on complex spiking be included.

3) Data clarification. One reviewer thought the interpretation of the data would predict a transience of CR3. Please clarify the predictions (and results) regarding persistence/transience of the CRs.

4) Placing the work in context: Questions were raised regarding how this work interfaces with existing models of cerebellar function (inverse model, forward model, etc.). Please discuss.

General assessments from reviewers:

Reviewer #1:

The question of how the brain learns to produce a sequence of actions is poorly understood. In this interesting and thorough paper, Mauk and colleagues present the case that the cerebellum is a key structure that chains the sequence of actions by using feedback from one action to produce the next action. I found the paper quite convincing, with a logical sequence of experiments that build on each other. Overall, the paper presents behavioral and neural evidence that learning of a sequence can be accomplished by the cerebellum through chaining of feedback from individual movements.

In this paper, CS was via mossy fiber stimulation, lasting 500ms, at the end of which a US was presented (electrical stimulation of the skin near the eye). Once the CS-US association was learned, the animal was trained to produce a second response CR2 without a CS. If eyelid response to the CS was less than 50% eye closure, US came as usual (I will label that as US1). If eyelid response was greater, US came 600ms after CS offset (I will label that as US2). Therefore, the training imposed an amplitude dependent response. If CR1 is large, US1 is omitted, but there will be a US2, and they learn to produce a CR2. If CR1 is small, US2 is presented. In this way, extinction of CR1 is prevented, while production of CR2 is trained. The interesting point is that there is no CS to drive learning of CR2, with the likely signal being production of CR1 itself.

The primary hypothesis was that CR2 was learned because of production of CR1. The idea then is that CS is associated with US, resulting in CR1. Then CR1 itself is associated with the US2, producing CR2. The most convincing evidence for me regarding this hypothesis is the control experiment in which extinction of CR1 results in extinction of CR2. They omitted US1 and always presented US2. They find that as CR1 declined, so did CR2, despite presence of US2. This is the best evidence for the hypothesis that production of CR1 is being linked to production of CR2. Without CR1, there is no CR2.

The results end with recordings from 42 P-cells, among which 16 were eyelid related. During sequence training, P-cell activity showed a decline during CS1, and then a second smaller decline before CS2. Modulation of P-cell response before CR2 appeared to be causal, as trials which did not have a CR2 did not show reduction in simple spikes. The timing of P-cell modulation for both CR1 and CR2 appears consistent with a role for the cerebellum in generating the CR2.

Overall, this is a sequence of logical experiments that begin with a curious finding and build toward an exciting and important new understanding of the cerebellum.

Reviewer #2:

The manuscript by Khilkevich, Zambrano, Richards, and Mauk presents the findings of a series of experiments that were designed to examine sequence learning within the cerebellum. The authors used eyelid conditioning procedures to train movement sequences in rabbits with mossy fiber stimulation as the CS (and a peripheral CS in one experiment). They were able to train a second conditioned response (CR) by making the delivery of a relatively late unconditioned stimulus (US) contingent on the amplitude of the initial CR. A series of behavioral experiments made a very convincing case that feedback from the first CR was causally related to the second CR. The authors also examined Purkinje cell correlates of the two CRs using the ipsilateral stimulation paradigm. As this group showed previously with a single CR paradigm, decreases in Purkinje cell simple spike activity correlated with the amplitude and time course of the eyelid CR. They found that these correlations were also very high for the second CR in the ipsilateral sequence learning paradigm. Thus, Purkinje cell activity appears to be causally related to the production of both CRs.

The design of the experiments is very clever, with superb control over the timing of the learning events and the ability to establish two distinct CRs, which has been a limitation of previous studies that trained animals to learn multiple eyelid CRs. The paper is well written and the figures provide the necessary data to evaluate the results.

Reviewer #3: In this manuscript, Khilkevich et al. test whether and how known mechanisms of associative cerebellar sensorimotor learning can be harnessed to generate motor sequences. Specifically, the authors test the fundamental question of whether feedback from a learned movement can provide a substrate for learning subsequent movements in a motor sequence. To address this question, they utilize a well-characterized form of cerebellar-dependent sensorimotor learning in rabbits that involves eyelid conditioning driven by electrical stimulation of cerebellar mossy fibers. This electrical stimulation approach has been demonstrated to produce classically conditioned eyelid responses that do not require a traditional sensory conditioned stimulus (CS), and has the powerful advantage of restricting inputs only to the cerebellum. Using this paradigm, the authors convincingly demonstrate the following key results:

1) Following initial eyelid conditioning, delivery of a delayed unconditioned stimulus (US) at time when mossy fiber stimulation cannot drive a conditioned response (CR) produces a second, late CR. This new form of learning has several important characteristics that are consistent with the mechanisms that have been previously described for cerebellar-dependent sensorimotor learning, including the protocol and timing for extinguishing learning.

2) Once a late CR is learned following repeated delivery of a late US to the ipsilateral eye, multiple sequential late CRs can be produced that have the same delay interval.

3) A delayed US delivered to the contralateral eye following ipsilateral conditioning can produce a learned contralateral CR, but not multiple sequential contralateral CRs.

4) Expression of late CRs, either ipsilateral or contralateral to the initially conditioned eyelid, has a strict trial-by-trial dependence on the occurrence and timing of initial CRs.

5) Purkinje cell simple spiking shows the same relationship between movement amplitude and timing for late CRs as for initial CRs.

Together, these findings (along with several carefully designed controls) strongly implicate a feedback signal to the cerebellum from the initial CR in generating later sequential CRs. Thus, this manuscript represents a profound and elegant demonstration that internally generated signals can be exploited to produce motor sequences in absence of continued external sensory cues. Moreover, by showing that the cerebellum is capable of generating and storing such motor sequences, this work provides a plausible explanation for why cerebellar patients have severe sequence learning deficits, and suggests that the cerebellum may be a key structure for establishing motor programs that require multiple well-timed movements.

In summary, this is an outstanding manuscript that moves the field forward by revealing crucial new features cerebellar learning (and motor learning more broadly) that will be of interest to a wide audience.

Major points:

1) Data presentation/statistical analysis:

A) During Control 2 experiments, CS1 as well as CS2 were associated with US1. Then during sequence training only CS1 was used. Occasionally, a probe trial was inserted where a CS2 would be presented. This elicited a CR1, but the question was whether presence of CR1 would be sufficient to be followed by CR2. You find that if and only if CS2 produced a CR1, then it was followed by a CR2 (occasionally CS2 did not produce a CR1, and in those cases there was also no CR2). This is a critical result, but the presentation of the data is inadequate. Your results should be shown as a conditional probability relationship, not correlation of likelihoods (Figure 5D). Figure 6 data are confusing and should be replaced with data that test for existence of a conditional probability (if CR1, then CR2). The control condition is the null hypothesis of independence of two events.

B) Similar to my comment above, correlation coefficients are inappropriate for quantifying temporal dependence of one event on another. You are asking the following question: given that CR1 and CR2 occurred, was the timing of CR2 conditioned on timing of CR1? The null hypothesis is that given that CR1 and CR2 occurred, the timing of CR2 was not conditioned on timing of CR1.

2) Complex spikes: Like all good papers, the work demonstrates a novel finding that naturally raises follow-up questions. The presentation of the simple spikes is critical for providing evidence that learning of CR2 appears to be similar to CR1. The data in Figure 8 for this question is convincing. However, because complex spikes have also been recorded, presentation of the simple spikes begs the question of what happened to the complex spikes. I request that another figure be added to show data on the complex spikes.

3) Data Clarification: In 36% of late acquisition sessions if there was a CR2, there was also a CR3, with appropriate relative timing. The problem is that unlike the condition for learning of CR2 where a US2 is present, there is no error to the cerebellum to encourage learning of CR3. Without an error signal, even when CR3 occurs, it should undergo extinction. Therefore, CR3 should be a transient part of the learning process that might develop as CR2 is learned, but then disappear without a US3. Is there data on this?

4) Placing work in context: The current paper does not attempt to relate the proposed sequence learning mechanism to theories of cerebellar contributions to movement that posit internal models. How does the proposed mechanism relate to theories of cerebellar control that posit an inverse model, forward model, or a combination within the cerebellum? Because the current sequence learning mechanism requires reinforcement of each component of the sequences, this type of learning mechanism is not obviously consistent with previous theories positing internal models. Thus, the authors need to clarify whether or not their view is consistent with the inverse and forward models of cerebellar control. The end of the discussion seems to imply an internal model, but what type of model and how it relates to other theories was not clear.

5) The Purkinje cell recordings demonstrate nice correlates of the two CRs generated with the ipsilateral paradigm, but these findings simply reinforce the previously established causal relationship between decreases in simple spike activity and CR dynamics. The key for making this project highly significant would be to identify the feedback signal and how it relates to the development of precisely timed changes in Purkinje cell activity.

---

## [Author Response]

Essential revisions:The reviewers all found the work to be interesting and convincing. The "General Assessments" from the reviewers are all included below for the authors' information and satisfaction. Since they are so favorable, we thought the authors would like to read them in their raw form.Major points:1) Data presentation/statistical analysis:A) During Control 2 experiments, CS1 as well as CS2 were associated with US1. Then during sequence training only CS1 was used. Occasionally, a probe trial was inserted where a CS2 would be presented. This elicited a CR1, but the question was whether presence of CR1 would be sufficient to be followed by CR2. You find that if and only if CS2 produced a CR1, then it was followed by a CR2 (occasionally CS2 did not produce a CR1, and in those cases there was also no CR2). This is a critical result, but the presentation of the data is inadequate. Your results should be shown as a conditional probability relationship, not correlation of likelihoods (Figure 5D).

We apologize for the confusion. In eyelid conditioning literature the terms “likelihood” and “probability” are often used interchangeably, referring to the fraction of trials where CRs were observed. We agree that “likelihood” has a quite distinct meaning in statistics and thus can be confusing to the reader. We therefore clarified the definition of CR probability in the text (subsection “Training an ipsilateral sequence of CRs”) and updated our terminology throughout the text and figures.

With that said, we believe that Figure 5C and G show the data exactly in the format the reviewer is asking. There, the probability of CR2 is shown for four conditions (CS1 trials with CR1, CS2 trials with CR1, CS2 nonCR1 trials and a control for spontaneous eyelid movement).

Results in Figure 5D and H are less critical and show an existence of correlation between first two conditions on different sessions (probability of CR2 on CR1 trials, elicited by CS2 (Y axis) or CS1 (X axis)). We updated and color-coded axis labels to make this transition clearer.

Figure 6 data are confusing and should be replaced with data that test for existence of a conditional probability (if CR1, then CR2). The control condition is the null hypothesis of independence of two events.

Similarly to the previous point, Figure 6E and F already show the probability of CR2 conditioned of CR1 amplitude. We again apologize for the confusion.

B) Similar to my comment above, correlation coefficients are inappropriate for quantifying temporal dependence of one event on another. You are asking the following question: given that CR1 and CR2 occurred, was the timing of CR2 conditioned on timing of CR1? The null hypothesis is that given that CR1 and CR2 occurred, the timing of CR2 was not conditioned on timing of CR1.

We thank the reviewer for this suggestion. We calculated distribution of time intervals between corresponding timing measures of CR1 and CR2 and, for comparison, the same distribution with permuted trials for CR1 data. If the timing of CR2 was conditioned on timing of CR1, the distribution with real inter-CR intervals should differ from the distribution after permutation. We show the real (appropriately colored) and shuffled (grey) distributions on the right of corresponding panels in Figure 7. We found a significant difference between distributions in all cases where the number of trials was sufficient (see Figure 7—source data 1).

We added the description of this new analysis in subsection “Prediction 4: Timing of CRs in a sequence should co-vary on a trial-by-trial basis”. We believe though that the correlation analysis is also informative and did not remove its results.

2) Complex spikes: Like all good papers, the work demonstrates a novel finding that naturally raises follow-up questions. The presentation of the simple spikes is critical for providing evidence that learning of CR2 appears to be similar to CR1. The data in Figure 8 for this question is convincing. However, because complex spikes have also been recorded, presentation of the simple spikes begs the question of what happened to the complex spikes. I request that another figure be added to show data on the complex spikes.

We appreciate the reviewer’s interest. We would like to explain though why we did not put complex spike data in the first place. While we put effort into separating complex spikes from simple spikes of PCs, our goal was simply to assess whether PC responded to US with complex spikes (qualifying for definition of eyelid PC) or not. For this purpose we did not necessary need to reliably recognize every complex spike. Accordingly, we utilized a previously described cluster-cutting procedure (Halverson et al., 2015), which relies on the consistency of complex spike waveform. This is true only to a degree and it is likely that in some cases we are not counting some complex spikes or over-counting in others, if late spikelets are recognized as separate events.

For these reasons we do not feel comfortable to perform a fine level of analysis on the complex spikes data that we have.

3) Data Clarification: In 36% of late acquisition sessions if there was a CR2, there was also a CR3, with appropriate relative timing. The problem is that unlike the condition for learning of CR2 where a US2 is present, there is no error to the cerebellum to encourage learning of CR3. Without an error signal, even when CR3 occurs, it should undergo extinction. Therefore, CR3 should be a transient part of the learning process that might develop as CR2 is learned, but then disappear without a US3. Is there data on this?

We improved the explanation in subsection “Training an ipsilateral sequence of CRs” why CR3 do not extinguish (as long as CR2 do not). We also put into notation “FS_L_” and “FS_R_” to better distinguish between FS from left and right eyelid CRs in contralateral sequence.

We see the fact that CR3 remains without a direct reinforcement as evidence that the effective “CS” is the same for both CR2 and CR3 in ipsilateral sequence. Because of this, reinforcement of CR2 on a portion of trials is sufficient to prevent both CR2 and CR3 from extinguishing.

4) Placing work in context: The current paper does not attempt to relate the proposed sequence learning mechanism to theories of cerebellar contributions to movement that posit internal models. How does the proposed mechanism relate to theories of cerebellar control that posit an inverse model, forward model, or a combination within the cerebellum? Because the current sequence learning mechanism requires reinforcement of each component of the sequences, this type of learning mechanism is not obviously consistent with previous theories positing internal models. Thus, the authors need to clarify whether or not their view is consistent with the inverse and forward models of cerebellar control. The end of the discussion seems to imply an internal model, but what type of model and how it relates to other theories was not clear.

We agree that the internal model framework is one of the prominent theories of motor control and cerebellar contribution to it. We think though that eyelid conditioning in general is ill-suited for studying the specifics of internal model involved. In our case, the cerebellum does not only learn a prediction but also serves as a controller that sends a motor signal to the eyelid plant.

We thus chose to place our sequence learning mechanism in context of general dynamical interaction between the cerebellum and cortical areas, being agnostic to the model that describes the cerebellar function.

5) The Purkinje cell recordings demonstrate nice correlates of the two CRs generated with the ipsilateral paradigm, but these findings simply reinforce the previously established causal relationship between decreases in simple spike activity and CR dynamics. The key for making this project highly significant would be to identify the feedback signal and how it relates to the development of precisely timed changes in Purkinje cell activity.

We agree about the importance of identifying the feedback signal route and hope that future studies will address this exciting question. Since multiple possible routes exist, the answer might not be singular and can depend on relative laterality and timing of responses in the sequence.

We expanded the section in Discussion that addressed the above question and added a possible feedback route from red nucleus.